# Prompt-Based Monte-Carlo Tree Search for Goal-oriented Dialogue Policy Planning

**Xiao Yu, Maximillian Chen, Zhou Yu**

Department of Computer Science, Columbia University, New York, NY

{xy2437, zy2461}@columbia.edu, maxchen@cs.columbia.edu

## Abstract

Planning for goal-oriented dialogue often requires simulating future dialogue interactions and estimating task progress. Many approaches thus consider training neural networks to perform look-ahead search algorithms such as A* search and Monte Carlo Tree Search (MCTS). However, this training often requires abundant annotated data, which creates challenges when faced with noisy annotations or low-resource settings. We introduce GDP-ZERO, an approach using Open-Loop MCTS to perform goal-oriented dialogue policy planning *without any model training*. GDP-ZERO prompts a large language model to act as a policy prior, value function, user simulator, and system model during the tree search. We evaluate GDP-ZERO on the goal-oriented task PersuasionForGood, and find that its responses are preferred over ChatGPT up to 59.32% of the time, and are rated more persuasive than ChatGPT during interactive evaluations[1].

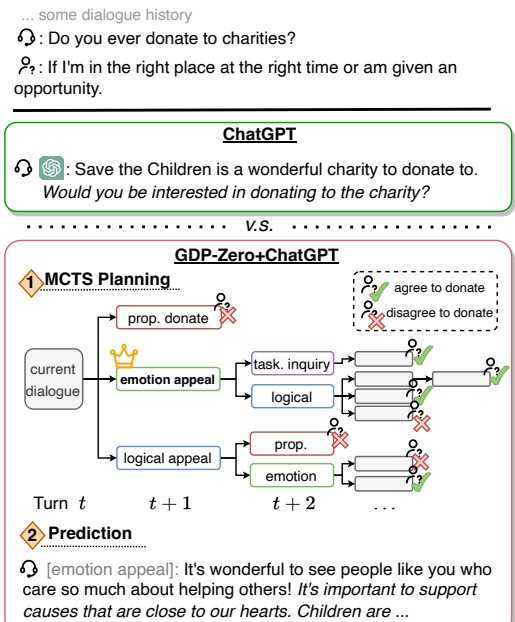

Figure 1: Using GDP-ZERO for persuasion with zero model training.

## 1 Introduction

In many goal-oriented conversation tasks, interacting parties must retake initiative (Allen et al., 1999) by executing conversational strategies to lead the conversation a desired outcome (e.g. successful negotiation (Lewis et al., 2017) or emotional support (Liu et al., 2021)). As such, it is imperative to have high-quality dialogue policy planners that can prescribe an "optimal" strategy at each turn of the dialogue (Levin et al., 1997; Zhang et al., 2020b; Liu and Lane, 2017; Liu et al., 2018).

Optimal policy planning is a difficult task. While in task-oriented settings (e.g. restaurant booking) there at least is objectivity with respect to successful planning, many goal-oriented tasks like persuasion are often subjective. For instance, in a persuasion task, individual persuaders might adopt different strategies, making it difficult to train or

evaluate a policy planner[2]. Moreover, "optimality" in these complex tasks may require expert domain knowledge (e.g., negotiation skills). This also makes collecting and annotating high-quality conversations difficult (Chen et al., 2023b).

In this work, we contribute a novel approach to **G**oal-oriented **D**ialogue **P**lanning with **Zero** training (GDP-ZERO). GDP-ZERO prompts a large language model (LLM) to perform planning by simulating future dialogue interactions (Figure 1), making it particularly suitable for tasks which would otherwise require high-quality conversations

---

[1]Code available at: https://github.com/jasonyux/GDPZero

[2]In our extensive initial experiments to build a policy planner for the PersuasionForGood (Wang et al., 2019) task, we found 1) training a supervised or reinforcement learning-based multi-class classifier for next-dialogue-act-prediction yields only 18-25% accuracy at best (including finetuning LLaMA (Touvron et al., 2023), RoBERTa (Liu et al., 2019), and prompting LLMs such as ChatGPT), and 2) converting the multiclass policy prediction task into a binary task of whether to "propose to donate" still yields only 74% accuracy.

and annotations. Unlike previous approaches, we treat policy planning as a stochastic game, and use prompting for every stage of an open-loop tree search. We evaluate GDP-ZERO on PersuasionFor-Good due to its difficult planning task (Wang et al., 2019), and find its responses are preferred over ChatGPT in both static and interactive evaluations.

## 2  Related Work

**Prompting Methods**  Few-shot dialogue techniques have a lot of advantages, including out-of-domain generalization (Zhao and Eskenazi, 2018; Mehri and Eskenazi, 2021) and difficult low resource settings with noisy annotations (Chen et al., 2023b). Recently, prompting LLMs has become the predominant approach to few-shot language tasks, and its applications in dialogue have received much attention. However, this has largely focused on dialogue response generation (e.g. Chen et al. (2023b); Liu and Kilicoglu (2023); Madotto et al. (2021); Liu et al. (2022)), conversation synthesis (e.g. Chen et al. (2023a); Kim et al. (2022); Bae et al. (2022)), and dialogue understanding (e.g. Yang et al. (2022); Gupta et al. (2022)). To date, prompting has not been used for policy planning.

**Dialogue Policy Planning**  Research on dialogue policy planning can be categorized into neural-focused and algorithmic-focused. Neural-focused approaches use annotated dialogues to train dedicated classifiers or value functions to predict the next dialogue acts without explicit look-ahead planning (Zhang et al., 2022a,b; Cao et al., 2020; Peng et al., 2018; Zhang et al., 2023). For many goal-oriented dialogues, however, both annotated strategies and dialogue responses can be sub-optimal/noisy, as different people can respond differently even given the same context.

To reduce the reliance on a labeled dataset, much work has also attempted to combine neural networks with search algorithms, such as A* search (Cheng et al., 2022) and tree search (Wang et al., 2020; Yang et al., 2021; Jang et al., 2020; Väth et al., 2023). However, these methods still require model training for dialogue simulation or value function estimation, and are therefore highly dependent on training data quality (Shi et al., 2019). For example, Jang et al. (2020) use MCTS for training an RNN-based policy model and Wang et al. (2020) train multiple neural networks for user simulation and value function estimation. Consequently, these methods can face difficulties during dialogue

simulation due to a) noisy data annotations causing sub-optimally trained generation models, and b) inaccurate responses generated at turn $i$ compounding errors for simulations at turns $> i$.

## 3  Method

In this work, we introduce GDP-ZERO, an algorithm-focused dialogue policy planner for goal-oriented dialogue tasks like persuasion. GDP-ZERO uses zero model training and instead performs Open-Loop MCTS at decision time by prompting an LLM to simulate user and system response, evaluate current task progress, and predict a prior next dialogue act. Building on findings from Chen et al. (2023b), our approach has two main differences from existing policy planning work: we use few-shot prompting to bypass the need for model training on noisy data, and we use Open-Loop MCTS to reduce compounding simulation errors by continuously re-generating system and user responses during the tree search.

### 3.1  Problem Definition

To introduce tree search methods for dialogue policy planning, we first formulate planning as a Markov Decision Process (MDP). A $t$ turn dialogue between a user and a system can be represented as:

$$h = (a_0^{\text{sys}}, u_1^{\text{sys}}, u_1^{\text{usr}}, ..., a_{t-1}^{\text{sys}}, u_t^{\text{sys}}, u_t^{\text{usr}})$$

where $a_i^{\text{sys}}$ is the system's dialogue act at turn $i$, $u_i^{\text{sys}}$ is the system's response, and $u_i^{\text{usr}}$ is the user's utterance at turn $i$. Similar to Yang et al. (2021) and Wang et al. (2020), we define the task of planning the next $a^{\text{sys}}$ as an MDP problem $\langle \mathcal{S}, \mathcal{A}, \mathcal{R}, \mathcal{P}, \gamma \rangle$. The dialogue act of the system $a_i^{\text{sys}}$ represents an action $a_i \in \mathcal{A}$ at a turn $i$, and the corresponding dialogue history up to the $i$-th turn $s_i = (a_0, u_1^{\text{sys}}, u_1^{\text{usr}}, ..., a_{i-1}, u_i^{\text{sys}}, u_i^{\text{usr}})$ represents a state $s_i \in \mathcal{S}$. A reward function $\mathcal{R}(s, a)$ represents the likelihood of a desired conversational outcome, such as persuading a user to donate to a charity. The transition function $\mathcal{P} : \mathcal{S} \times \mathcal{A} \to \mathcal{S}$ represents the probability of transitioning from a dialogue state $s_i$ to state $s_{i+1}$ after executing $a_i$ at a turn. Finally, $\gamma \in [0, 1)$ is the discount factor.

### 3.2  Dialogue Planning as a Stochastic MDP

In a typical MDP game like Go, much success has been achieved by employing closed-loop MCTS to plan for the next move/action $a$ (Silver et al., 2016, 2017). In general, MCTS algorithms improve the

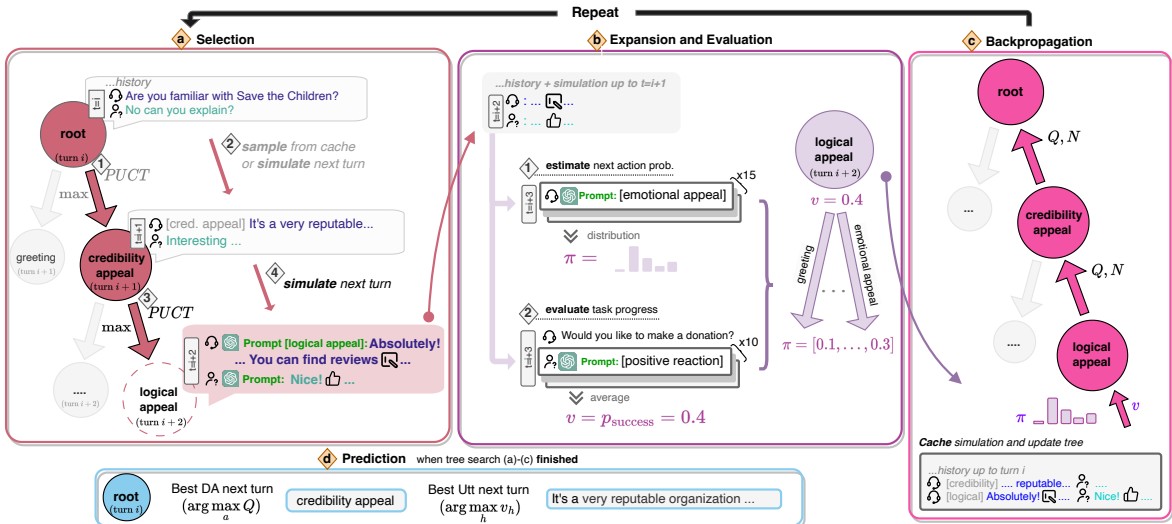

Figure 2: GDP-ZERO with ChatGPT backbone. During **Selection**, simulations are either sampled from cache or newly generated. During **Expansion and Evaluation**, we prompt ChatGPT for prior policy $\pi$ and value estimation.

actions proposed by an original policy model $\pi_\theta$ by building a search tree that contain simulated outcomes of many potentially high-rewarding actions, and returning the best action according to the simulation/evaluation results. Specifically, MCTS repeatedly 1) explores a new action or exploits a successful action $a$ proposed by $\pi_\theta$ in a state $s$; 2) simulates the action outcome $s' \leftarrow \mathcal{P}(s,a)$; 3) evaluates the action's quality; and 4) updates its estimate $Q(s,a)$ for that action. At the end of the search, MCTS produces probabilities $\pi \propto N(s,a)^{1/\tau}$ for playing each action using the exponentiated visit count, which usually suggests much stronger moves than using the raw policy $\pi_\theta$ (Howard, 1960; Sutton and Barto, 2018; Silver et al., 2017).

However, in simulating *dialogue* interactions during tree search, generating a slightly improbable system or user response for state $s'$ and storing it in a search tree could lead to a large compounding error for the rest of the subtree from $s'$ (Wang et al. (2020)). This is because the state space representing all possible responses is large, and dialogue responses are diverse. This makes it difficult to accurately simulate the most probable next dialogue state $s'$ given an $(s,a)$ pair from the previous turn. We thus treat dialogue policy planning as a stochastic MDP, where the simulated next state $s' \leftarrow \mathcal{P}(s,a)$ is drawn from a large unknown distribution and might not be representative of the most probable $s'$ (Perez Liebana et al., 2015). Unlike previous usages of (closed-loop) MCTS for dialogue which consider a deterministic transition (Wang et al., 2020; Yang et al., 2021), this formulation re-

quires potentially different $s'$ to be returned given the same dialogue context $s$ and system action $a$.

### 3.3 GDP-ZERO

To solve this stochastic problem, we base our algorithm on Open-Loop MCTS (Weber, 2010; Perez Liebana et al., 2015), a variant of MCTS where each tree node $s_i^{tr} = (a_0, ..., a_i)$ represents the sequence of dialogue *actions* to reach a dialogue turn $i$. Instead of using system/user utterances to represent a tree node, this design forces an algorithm to (re)generate the corresponding system and user utterances when traversing the tree (see Figure 2). Over time, a tree node $s^{tr}$ stores statistics derived from executing the sequence of dialogue actions (DA) *without* relying on some specific instance of user/system utterances, which could cause errors to propagate into future simulations. Given a dialogue state $s_0$, GDP-ZERO searches for the next best action by iteratively performing action **selection**, search tree **expansion**, action **evaluation**, and **backpropagation** to update tree statistics. After $n$ iterations, GDP-ZERO outputs a **prediction** for the next best action for $s_0$. GDP-ZERO is detailed in Figure 2 and Appendix A. Below we describe each stage of the algorithm.

**Selection**  Given a tree state $s^{tr}$, the action $a^*$ with the highest Predictor Upper Confidence Tree Bound (PUCT) (Silver et al., 2017; Rosin, 2011) is selected to traverse the tree:

$$\text{PUCT}(s^{tr}, a) = Q(s^{tr}, a) + c_p \frac{\sqrt{\sum_a N(s^{tr}, a)}}{1 + N(s^{tr}, a)},$$

where $N$ records the number of times a $(s^{tr}, a)$ pair has been visited, and $c_p$ is a hyperparameter controlling exploration. Since future simulations require a specific dialogue history, we either sample from the node's simulation cache if its size reached $k$, or generate a new simulation based on selected dialogue history $h^{tr}$ by prompting (Appendix B). We repeat this process until $s^{tr}$ becomes leaf node.

**Expansion** Once a leaf node is reached, we treat a LLM $\mathcal{M}_\theta$ as a prior policy by prompting it to generate a distribution of next dialogue acts. This is done by sampling $\mathcal{M}_\theta$ at temperature $\tau = 1.0$ for $m$ times, and converting the sampled DAs into a distribution (see Appendix A). Finally, each DA is also initialized with $Q(s^{tr}, \cdot) = Q_0$, a hyperparameter controlling exploration.

**Evaluation** We model the value of a state $v(s^{tr})$ by the probability that its dialogue context $h^{tr}$ can lead to task success. In a persuasion task to convince a user to donate to a charity, this can be achieved by appending the utterance "Would you like to make a donation?" to the context, and prompting an LLM $l$ times to simulate the user's current inclination to donation (Appendix A).

**Backpropagation** At the end of each search, we first store any newly simulated histories $h^{tr}$ in a cache for each $s^{tr}$. Then, we update the statistics of all nodes along the search path:

$$N(s^{tr}, a) \leftarrow N(s^{tr}, a) + 1 \qquad (1)$$
$$Q(s^{tr}, a) \leftarrow Q(s^{tr}, a) + \Delta Q(s^{tr}, a), \qquad (2)$$

where $\Delta Q(s^{tr}, a) = \frac{v(s^{tr}) - Q(s^{tr}, a)}{N(s^{tr}, a)}$. We also store a value $v_h$ estimated for each simulated history $h^{tr}$ that resulted in $v(s^{tr})$ by a running average:

$$v_h(h^{tr}) \leftarrow \frac{v_h(h^{tr}) \times N_h(h^{tr}) + v(s^{tr})}{N_h(h^{tr}) + 1}, \qquad (3)$$

with $N_h(\cdot)$ stores the number of times $h^{tr}$ is visited.

**Prediction** After all simulations are completed, we select the optimal action $a^* = \arg\max_a N(s_0^{tr}, a)$ based on the number of times an action has been visited, where $s_0^{tr}$ is the root node of the tree. To avoid generating a response using $\mathcal{M}_\theta$ on $a^*$ again, we also extract from cache the best system utterance associated with executing action $a^*$ at state $s_0^{tr}$. We use $v_h$ computed during the simulations, and return the utterance with the highest estimated value:

$$h_*^{tr} = \arg\max_{h_{a^*}^{tr}} v_h(h_{a^*}^{tr}),$$

where $h_{a^*}^{tr}$ denote any history $h^{tr}$ that played action $a^*$ after the root state $s_0^{tr}$. We call this process "*response selection*".

## 4 Experiments

We evaluate GDP-ZERO in the context of PersuasionForGood (P4G; Wang et al. (2019)), a dataset with 300 annotated dialogues where a "persuader" attempts to persuade a "persuadee" to donate to a charity called Save the Children (Appendix E). Due to the subjective nature of persuasion, it is not clear whether the annotated conversations reflect optimal persuasion strategies (Chen et al., 2022). This makes training and evaluating a policy planner challenging, since different persuaders likely adopt different strategies (Section 1). Motivated by this challenge, we employ GDP-ZERO to perform decision-time planning based on dialogue simulations, and focus on evaluating our approach's end-to-end performance in achieving the desired task outcome: successfully persuading a user to donate to Save the Children.

### 4.1 Static Evaluation

We use ChatGPT[3] (OpenAI, 2022) as the generation backbone of GDP-ZERO, which has become accepted as one of the most coherent general-purpose LLM (e.g. Liu et al. (2023b); Guo et al. (2023)). We take the first 20 dialogues from P4G, and produce 154 turns for evaluation. For each turn, we compare the response generated with and without GDP-ZERO for planning. Then, we prompted ChatGPT to choose[4] which generated response is more persuasive (Appendix F).

In Table 1, we found that generative approaches using ChatGPT were preferred over human ground truth responses about 90% of the time, with the highest score achieved by GDP-ZERO. In Table 2, we show that responses generated after GDP-ZERO planning were preferred in up to 59.32% of comparisons. We also observe increasing preference for GDP-ZERO when the number of simulations $n$ increases. Finally, we find changing $k, Q_0$ (controls

---

[3]Version "gpt-3.5-turbo" as of 2023 April

[4]ChatGPT can outperform crowdworkers on many annotation and evaluation tasks (e.g. Gilardi et al. (2023); He et al. (2023); Pan et al. (2023); Wang et al. (2021); Liu et al. (2023a))

| Algorithm | $n$ | $k$ | $Q_0$ | Win Rate v. Ground Truth |
|---|---|---|---|---|
| Prompting | - | - | - | $88.84 \pm 0.75\%$ |
| GDP-Zero | 5 | 3 | 0.00 | $87.22 \pm 0.61\%$ |
| GDP-Zero | 10 | 3 | 0.00 | $\mathbf{90.69 \pm 1.60\%}$ |
| GDP-Zero | 20 | 3 | 0.00 | $88.86 \pm 1.24\%$ |
| GDP-Zero | 50 | 3 | 0.00 | $89.82 \pm 1.10\%$ |

Table 1: Static evaluation with ChatGPT as backbone and judge. Results are $\mu \pm \sigma$ repeated over three runs.

| GDP-Zero (ChatGPT) | | | | Win Rate v. ChatGPT |
|---|---|---|---|---|
| $n$ | $k$ | $Q_0$ | Run Time | |
| 5 | 3 | 0.00 | 18s | $50.65 \pm 3.31\%$ |
| 10 | 3 | 0.00 | 36s | $50.86 \pm 1.10\%$ |
| 20 | 3 | 0.00 | 75s | $53.24 \pm 1.91\%$ |
| 50 | 3 | 0.00 | 740s | $\mathbf{59.32 \pm 1.84\%}$ |
| 10 | 1 | 0.00 | 16s | $49.57 \pm 2.01\%$ |
| 10 | 2 | 0.00 | 29s | $51.30 \pm 1.59\%$ |
| 10 | 3 | 0.25 | 36s | $57.79 \pm 2.95\%$ |
| 10 | 3 | 0.50 | 36s | $53.03 \pm 2.00\%$ |

Table 2: Static evaluation ChatGPT as backbone and judge. Runner-up is colored in gray. Results are $\mu \pm \sigma$ repeated over three runs.

| The chatbot... | RAP | ChatGPT | GDP-Zero |
|---|---|---|---|
| donation prob. ↑ | $0.72\pm0.38$ | $0.73\pm0.38$ | $\mathbf{0.79\pm0.37}$ |
| inc. donation intent ↑ | $4.08\pm0.68$ | $3.77\pm0.90$ | $\mathbf{4.30\pm0.71}$** |
| strong argument ↑ | $3.89\pm0.97$ | $3.91\pm0.99$ | $\mathbf{4.28\pm0.74}$* |
| is convincing ↑ | $4.11\pm0.74$ | $4.10\pm0.70$ | $\mathbf{4.38\pm0.66}$** |
| diverse strategy ↑ | $\mathbf{3.98\pm0.80}$ | $3.83\pm1.03$ | $3.95\pm0.82$ |
| is manipulative ↓ | $2.64\pm1.36$ | $2.96\pm1.38$ | $\mathbf{2.29\pm1.33}$** |
| is natural ↑ | $4.25\pm0.68$ | $4.03\pm0.65$ | $\mathbf{4.38\pm0.62}$** |
| is relevant ↑ | $\mathbf{4.64\pm0.54}$* | $4.31\pm0.86$ | $4.59\pm0.49$ |
| is coherent ↑ | $4.28\pm0.65$ | $4.06\pm0.89$ | $\mathbf{4.42\pm0.49}$* |

Table 3: Interactive evaluation using RAP, ChatGPT, and GDP-ZERO as planners, with ChatGPT used for response generation/backbone. Results are $\mu \pm \sigma$. All scores scaled to $[1, 5]$ except for "donation prob." being $[0, 1]$. * denotes $p < 0.05$, and ** denotes $p < 0.01$.

(Appendix H). Our study revealed that GDP-ZERO achieves the best performance across all metrics related to persuasiveness. We also found that RAP is highly rated for strategy diversity and relevance, indicating the benefit of using expert knowledge in planning. In Appendix D we provide some analysis of the generated dialogues and the resulting dialogue act distributions using different planners. In Appendix I we include some example dialogues.

### 4.3 Ablation Studies

To study how various components of GDP-ZERO helped improve task success, we conducted various ablation studies including 1) using Closed-Loop MCTS instead of the open-loop variant; 2) removing the "*response selection*" process; and 3) testing with both ChatGPT and Codex (Chen et al., 2021) as a backbone. Our experiments (see Appendix C) show that using Open-Loop MCTS contributed the most to making a response more persuasive, and that using the "*response selection*" precedure further improves performance. We also find that GDP-ZERO can improve upon both backbone models.

### 5 Conclusion

We propose GDP-ZERO, an algorithm to perform look-ahead policy planning with a large language model for goal-oriented dialogues. End-to-end evaluation shows that using the policy from GDP-ZERO surpasses previous state-of-the-art system (RAP) and direct prompting with state-of-the-art LLMs on the task of persuasion without any model training. Strong performance in the zero-data regime opens the possibility of future work building dialogue systems in more conversational tasks under data-scarce settings.

simulation diversity and exploration, respectively) can slightly improve performance (Appendix A).

However, we consider the possibility that a) ChatGPT is biased towards its own generated dialogues (Liu et al., 2023a), and b) it might not have a robust criteria of what constitutes *persuasiveness*. As such, we also conducted interactive evaluation.

### 4.2 Interactive Human Evaluation

We conducted interactive human evaluation using the LegoEval platform (Li et al., 2021) with crowdworkers on Amazon Mechanical Turk. We primarily sought to evaluate GDP-ZERO in an end-to-end chatbot against two competitive baselines. The first is prompting ChatGPT for generation without GDP-ZERO planning. The second follows Chen et al. (2023b) by using ChatGPT with RAP (Chen et al., 2022). To our knowledge, RAP is the current state-of-the-art system on P4G, using trained modules with fixed dialogue policies derived from expert knowledge. Recently, Chen et al. (2023b) showed that using an LLM (e.g. ChatGPT) as the response generation module for RAP further improved performance (see Appendix G for details).

After the conversation, we asked the crowdworkers to evaluate our system based on the criteria in Table 3. We collected 40 survey results for GDP-ZERO, 35 for ChatGPT, and 36 for RAP

## 6 Limitations

**When is using GDP-ZERO appropriate?** In this paper, we present GDP-ZERO, a general approach for close-domain dialogue policy planning at turn-level. However, in this work we only evaluated GDP-ZERO on P4G. This is because we believe simulation-based plannings would be most beneficial when the task 1) often requires long-horizon planning to be successful, 2) does not have "optimal" action annotations readily available for supervised learning, and 3) evaluation does not require study participants to follow fake hypothetical scenarios to conduct a conversation, as otherwise the outcome may be disingenuous. We thus believe tasks like persuasion are most suitable, where planning ahead is crucial to success and policy optimality from human demonstrations is extremely subjective. Additionally, such a user may or may not want to donate to the charity, and the task of the dialogue system is to try to persuade them. This is in contrast to other goal-oriented contexts like task-oriented dialogue (TOD), where strong policies can be directly learned due to TOD's mostly passive and objective nature (e.g. He et al. (2022)), and also to dialogues where a hypothetical scenario is required. For example, to design a fair and controlled study, a user may need to make up fake preferences/objectives about a certain product in CraiglistsBargains (He et al., 2018), or pretend that they need emotional support for some fixed set of issues in ESConv (Liu et al., 2021).

Additionally, while GDP-ZERO can be adapted to task-oriented contexts like Multi-Woz (Budzianowski et al., 2018), it may not necessarily be appropriate. Such task-oriented contexts often have hierarchical policies (e.g. "[hotel] [recommend] name price" and "[restaurant] [inform] food price area"), and adaptation to GDP-ZERO would require converting the hierarchy into a multi-label classification, resulting in a massive action space. We believe this could be very inefficient, and approaches such as building multiple search trees to perform high/low-level planning would be useful (Zhang et al., 2020a).

**Runtime** One important limitation for GDP-ZERO is runtime. The more exhaustive the tree search (e.g. increasing $n$ or $k$), the more likely the algorithm is able to find the optimal dialogue policy (Table 2). However, this comes at the cost of longer simulation time, which may affect the overall user experience, and accordingly, user perceptions of persuasiveness.

With OpenAI API's rate limit and LLM's inference speed, we restricted GDP-ZERO to plan on 7 dialogue acts in P4G, with $n = 10, k = 3$ for a simulation time of around 35 seconds during interactive evaluation. We believe methods to parallelize tree search (Chaslot et al., 2008) or to re-use part of the simulation subtrees could be helpful to speed up GDP-ZERO. We expect that as research with LLMs progresses, inference speed will continue to improve. In the short-term, one may bypass latency limitations by utilizing multiple accounts to parallelize API calls during simulation.

**Simulation Quality** GDP-ZERO prompts a LLM (e.g. ChatGPT) to perform dialogue simulation and value estimation. Despite LLM's strong few-shot performance on many tasks, issues with controllable generation can still create errors during simulation (e.g. generated system utterances might not match planned dialogue action). GDP-ZERO accounts for such errors by using an Open-Loop search with $k > 1$, but this increases simulation runtime. We believe this trade-off between simulation quality and runtime is also an important aspect for future work to consider.

**Using ChatGPT for static analysis** While ChatGPT is shown to outperform crowdworkers on many annotation and evaluation tasks (e.g. Gilardi et al. (2023); He et al. (2023); Pan et al. (2023); Wang et al. (2021); Liu et al. (2023a)), recent work has been finding that LLMs like ChatGPT may favor responses with a higher number of unique tokens (Wang et al., 2023). As such, we analyzed responses between GDP-ZERO ($n = 20, k = 3, Q_0 = 0$) and ChatGPT, and found that there is indeed some positive correlation ($r = 0.29, p < 0.001$) between the number of words in a response and ChatGPT's preference for persuasiveness. However, it is not clear whether this is a limitation in our setting. Our setting of evaluating persuasiveness is not quite the same as the typical context of considering direct preference; there are many situations where longer responses which correlate with persuasiveness (for instance, an aptly used logical appeal). As such, we conducted interactive evaluation with crowdworkers in Section 4.2 and find that GDP-ZERO generated responses are indeed rated as more persuasive.

## 7 Ethical Considerations

Our work describes an algorithm to perform dialogue policy planning for goal-oriented tasks without any model training. It is aimed at making future dialogue systems to build, and also better at helping users/systems achieve their tasks/goals.

**Potential Abuses** Generally, while most algorithms are not designed for unethical usage, there is often potential for abuse in their applications. In our experiments with PersuasionForGood (Wang et al., 2019), we apply GDP-ZERO on the goal of increasing users' intention to donate to a charity. However, because GDP-ZERO is fundamentally goal-agnostic, it is possible to use them for unethical tasks, such as scamming. We do not condone the use of GDP-ZERO for any unlawful or morally unjust purposes.

**Interactive Human Evaluation** In this study, we conducted interactive human evaluation using crowdworkers on the Amazon Mechanical Turk platform. All crowdworkers were informed that they were speaking with a chatbot. All study participants were paid at a rate of $15 per hour. Our study has received IRB approval.

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

## A  Additional details on GDP-Zero

We describe the details of GDP-Zero in Algorithm 1. Similar to other MCTS algorithms, GDP-Zero performs simulation based on four stages, selection, expansion, evaluation, and backpropagation, and finally predicts an action based on the simulations. Different from existing implementations, GDP-Zero performs Open-Loop search using *only a generative LLM $\mathcal{M}_\theta$*, by prompting it to do dialogue simulation, value function estimation, and prior policy estimation (see Appendix B for prompting details and examples).

GDP-Zero requires a generative LLM $\mathcal{M}_\theta$ as a backbone model, and takes in a dialogue history $h_i$ at turn $i$ as input. Given some fixed dialogue action space $\mathcal{A}$ (see appendix E for P4G), GDP-Zero builds a search tree after $n$ simulations. For each state, GDP-Zero keeps a cache of size $k$ storing newly generated user and system utterances. We use $c_p = 1.0$, and $Q_0 = \{0.0, 0.25, 0.5\}$ to promote exploration (see Table 2).

## B  Prompting Details on P4G

For P4G, we used the same one-shot example for all cases, while dynamically changing the representation for each operation.

**System response generation.** Following Chen et al. (2023b), we include the natural language form of a planned dialogue action (Table A4) in the prompt to perform conditional generation. We present an example in Table A7.

**User response generation.** We swap the user and the system role for this task, and prompt the LLM to act as a user simulator. We present an example in Table A8.

**Value function estimation.** To evaluate the user's inclination to donate at a given state, we first append the turn "Persuader: Would you be interested in donating to Save the Children?" to the dialogue history, and then prompt the LLM at temperature $\tau = 1.1$ to sample the user's response for $l = 10$ times. We define "no donation"=-1.0, "negative reaction"=-0.5, "neutral"=0.0, "positive reaction"=0.5, and "donation"=1.0, and then convert the sampled responses to a score between -1.0 and 1.0. We present an example in Table A9.

**Prior policy estimation.** We treat the backbone LLM as a prior policy, and prompt it to generate the next dialogue action at temperature $\tau = 1.0$ for 15

times to simulate its policy distribution. Finally, to promote the diversity of the generated dialogue actions during GDP-Zero tree search, we use add-1 smoothing to convert the generated dialogue actions to a probability distribution. We present an example in Table A10.

---

**Algorithm 1** GDP-Zero $(M_\theta)$

---

**Require:** generative LLM $\mathcal{M}_\theta$
**Require:** dialogue history $h_i$ until turn $i$
**Require:** dialogue action space $a \in \mathcal{A}$
**Require:** hyperparameter $n, k, c_p, Q_0$

1: Repeat for $n$ searches:
2:   initialize root node $s_i^{tr}$, $H(s_i^{tr}) \leftarrow \{h_i\}$
3:   $s^{tr} \leftarrow s_i^{tr}$
4:   *// selection*
5:   **while** $s^{tr}$ is not a leaf node **do**
6:     $a' \leftarrow \arg\max_a \text{PUCT}(s^{tr}, a; c_p)$
7:     $h^{tr} \leftarrow \text{sample}(H(s^{tr}))$
8:     $s^{tr} \leftarrow s^{tr} \cup a'$
9:     **if** $\text{len}(H(s^{tr})) < k$ **then**
10:       generate $h_{\text{new}} \leftarrow \mathcal{M}_\theta(h^{tr} \circ a')$
11:       save $H(s^{tr}) \leftarrow H(s^{tr}) \cup h_{\text{new}}$
12:     **end if**
13:   **end while**
14:   $h^{tr} \leftarrow \text{sample}(H(s^{tr}))$
15:   *// expansion*
16:   generate $p(a|s^{tr}) \leftarrow \mathcal{M}_\theta(h^{tr})$
17:   $s^{tr}.p \leftarrow p(a|s^{tr}), s^{tr}.Q \leftarrow Q_0, s^{tr}.N = 0$
18:   *// evaluation*
19:   generate $v(s^{tr}) \leftarrow \mathcal{M}_\theta(h^{tr})$
20:   *// backpropagation*
21:   **while** $s^{tr} \neq s_i^{tr}$ **do**
22:     update $v_h(h^{tr})$ with eq. (3)
23:     save simulation $H(s^{tr}) \leftarrow H(s^{tr}) \cup h^{tr}$
24:     $(s^{tr}, a) \leftarrow$ back to parent of $s^{tr}$
25:     update $Q(s^{tr}, a), N(s^{tr}, a)$ with eq. (1)
26:   **end while**
27: *// prediction after $n$ simulations*
28: $a^* \leftarrow \arg\max_a N(s_i^{tr}, a)$
29: $s_*^{tr} \leftarrow s_i^{tr} \cup a^*$
30: $u^{sys*} \leftarrow \arg\max_{u^{sys}} v_h(H(s_*^{tr}))$
31: **return** $a^*, u^{sys*}$

---

## C  Ablation Studies

In Table A1 and Table A2, we conduct ablations to study how each component in GDP-Zero affect its performance. We use the same 20 dialogues from our static evaluation (Section 4.1), and test

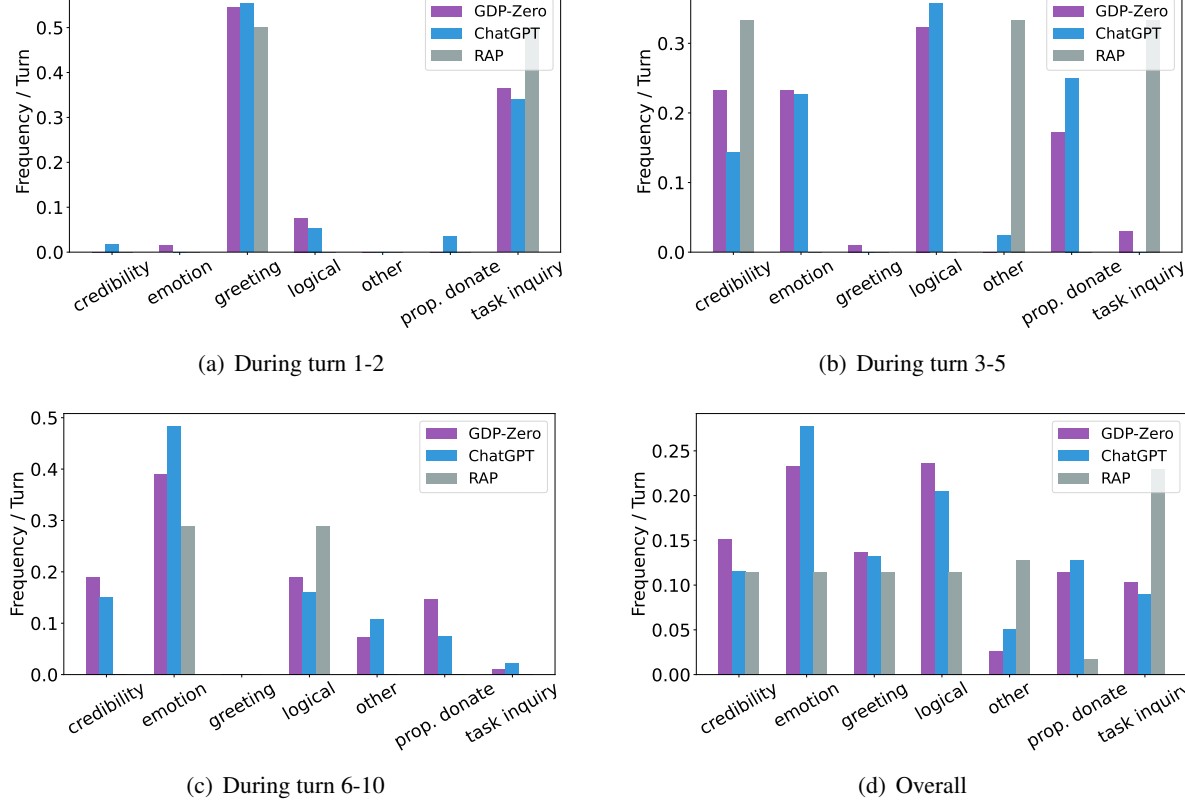

(a) During turn 1-2

(b) During turn 3-5

(c) During turn 6-10

(d) Overall

Figure A1: Distribution of planned dialogue actions for each planner during interactive evaluations.

GDP-Zero with ChatGPT (Table A2) and Codex[5] (Table A1) as the generation backbone. Similar to Section 4.1, we use ChatGPT to evaluate the persuasiveness of the generated responses.

In all runs, we use $n = 20, c_p = 1, Q_0 = 0, k = 3$ for GDP-ZERO, when applicable. Table A1 and Table A2 shows that GDP-ZERO improves the policy proposed by the raw backbone model, and also that of a Closed-Loop MCTS implementation (denoted as "w/o OpenLoop") and without the "response selection" process (denoted as "w/o Resp.Select"). This indicates that using the open-loop variant and the proposed "*response selection*" process are beneficial for GDP-ZERO to improve dialogue planning.

## D   Analysis of GDP-ZERO Dialogues

In our experiments, we utilized GDP-ZERO to generate persuasive responses to convince a persuadee to donate to a charity. In this section, we provide an analysis of how and when GDP-ZERO impacts persuasiveness.

| Algorithm | Backbone | Win Rate v. GT |
|---|---|---|
| Prompting | Codex | $38.09 \pm 2.00\%$ |
| GDP-Zero | Codex | $\mathbf{45.46 \pm 2.95\%}$ |
| w/o OpenLoop | Codex | $39.16 \pm 3.42\%$ |
| w/o Resp.Select | Codex | $40.80 \pm 1.47\%$ |

Table A1: Static evaluation using the first 20 dialogues of P4G with ChatGPT as judge. GT refers to Ground Truth. Results are $\mu \pm \sigma$ repeated over three runs.

| Algorithm | Backbone | Win Rate v. GT |
|---|---|---|
| Prompting | ChatGPT | $87.21 \pm 0.60\%$ |
| GDP-Zero | ChatGPT | $\mathbf{91.13 \pm 0.30\%}$ |
| w/o OpenLoop | ChatGPT | $88.09 \pm 0.81\%$ |
| w/o Resp.Select | ChatGPT | $91.03 \pm 0.75\%$ |

Table A2: Static evaluation using the first 20 dialogues of P4G with ChatGPT as judge. GT refers to Ground Truth. Results are $\mu \pm \sigma$ repeated over three runs. Since ChatGPT generations are typically long, we use the first 3 sentences of each generation in this evaluation.

**Static evaluation.**   Our static evaluation (Section 4.1) uses fixed dialogue contexts from the P4G dataset (Wang et al., 2019), and every turn of the conversation is already annotated with dialogue

---

[5]At the time of writing, Codex (Chen et al., 2021) was freely accessible from the OpenAI API.

actions. We thus utilize those annotations and investigate *when* GDP-ZERO is producing more persuasive responses. For this analysis GDP-ZERO used $n = 10, k = 3, Q_0 = 0.25$, which is the same configuration as in our interactive evaluation (Section 4.2, Appendix G). We found that GDP-ZERO is rated as more persuasive 70.59% of the time when the user at the previous turn showed negative emotions (i.e. "negative-reaction-to-donation" or "negative-to-inquiry"), and 59.26% of the time when the user showed positive emotions (i.e. "positive-reaction-to-donation" and "positive-to-inquiry"). We believe this implies that GDP-ZERO's simulations are most beneficial in persuading users who is having little intent to donation. This is because look-ahead planning may, for instance, help ensure a response has covered all of the user's concerns regarding donation. We present an example in Table A6.

**Interactive evaluation.** Since each interactive dialogue is unlabeled, here we focus on analyzing the known dialogue actions generated by the different planners during those conversations. In Figure A1 we present the distribution of planned dialogue actions produced by RAP, ChatGPT, and GDP-ZERO. In general, the planned dialogue actions using ChatGPT and GDP-ZERO are unevenly distributed across different stages of the dialogue. Across different times of the conversation, ChatGPT and GDP-ZERO shared the most frequent DA at each stage is: "greeting" during turns 1-2, "logical appeal" during turns 3-5, and "emotion appeal" during turn 6-10. However, during turns 3-5 GDP-ZERO had a relatively even preference between "credibility appeal", "emotion appeal", and "logical appeal", while ChatGPT strongly preferred "logical appeal." Additionally, we find that throughout all turns ChatGPT prefers to use "emotion appeal" over other dialogue actions, while GDP-ZERO balances between "emotion appeal" and "logical appeal," and RAP prefers "task related inquiry." It is also worth noting that GDP-ZERO performs "proposition of donation" less frequently at turn 1-5 than ChatGPT, and more frequently at turns 6-10. This indicates GDP-ZERO potentially adopting a more "conservative" strategy than ChatGPT overall. GDP-ZERO focuses on building up persuasive arguments and is less likely to select "proposition of donation" at early stages of the conversation in order to avoid over eargly propositions, which could lead to persuasion failure (also see

Table A12).

# E GDP-ZERO Setup on P4G

PersuasionForGood (P4G) is annotated with 10 persuasion strategies and 7 important non-persuasive strategies (see Table A3). However, since P4G is collected from human-human interaction, with both the "persuader" and the "persuadee" possibly donating to the charity, some of the dialogue actions are unsuitable when the "persuader" is a chatbot (e.g. self-modeling and personal story). We therefore choose a subset of dialogue actions to plan, by picking 4 frequent persuasive strategies suitable for chatbots, and 3 non-persuasive strategies including "other" to enable the chatbot to deal with unaccounted situations. We present the chosen dialogue actions and their prompts for LLM in Table A4.

| Dialogue Act | Category | Amount |
|---|---|---|
| Logical appeal | Strategy | 325 |
| Emotion appeal | Strategy | 237 |
| Credibility appeal | Strategy | 779 |
| Foot-in-the-door | Strategy | 134 |
| Self-modeling | Strategy | 150 |
| Personal story | Strategy | 91 |
| Donation information | Strategy | 362 |
| Source-related inquiry | Strategy | 167 |
| Task-related inquiry | Strategy | 180 |
| Personal-related inquiry | Strategy | 151 |
| Greeting | Non-Strategy | - |
| Proposition of donation | Non-Strategy | - |
| Proposition of amount | Non-Strategy | - |
| Proposition of confirmation | Non-Strategy | - |
| Proposition of more donat. | Non-Strategy | - |
| Experience affirmation | Non-Strategy | - |
| Thank | Non-Strategy | - |

Table A3: Annotated dialogue actions in the P4G dataset.

# F Additional details on static evaluation

In our static evaluation, we prompt ChatGPT to choose which generated response is better (e.g. with and without GDP-ZERO planning). Given two responses $u_a$ and $u_b$, we ask ChatGPT "Which of the following responses can better help the Persuader convince the Persuadee to donate to Save the Children? Why? A: $u_a$, B: $u_b$, C: Can't tell." after providing the relevant task context and dialogue history (see Table A5). For every evaluation, we sample the result 5 times and perform a majority vote. Interestingly, we find that ChatGPT skewed towards choosing option A, preferred choosing A

| Dialogue Act | Category | Natural Language Form |
|---|---|---|
| Logical Appeal | Strategy | The Persuader use of reasoning and evidence to convince the Persuadee. |
| Emotion Appeal | Strategy | The Persuader uses an emotion appeal to convince the Persuadee. |
| Credibility Appeal | Strategy | The Persuader establishes credibility of Save the Children by citing its impact. |
| Task Related Inquiry | Strategy | The Persuader asks about the Persuadee's knowledge or opinion related to Save the Children. |
| Proposition of Donation | Non-Strategy | The Persuader asks if the Persuadee would like to make a small donation. |
| Greeting | Non-Strategy | The Persuader greets the Persuadee. |
| Other | Non-Strategy | The Persuader responds to the Persuadee without using any persuasive strategy. |

Table A4: Mapping of persuader dialogue actions to natural language prompts used for prompting LLMs.

for 95.45% when $u_a = u_b$. We therefore randomly swap option A and B during all of our evaluations.

## G   Additional details on interactive study

In our interactive evaluation, we compare the rule-based planner from RAP, ChatGPT, and GDP-ZERO in an end-to-end chatbot for the persuasion task.

**RAP**   we use the rule-based planner derived from expert knowledge from RAP (Chen et al., 2022), which produces a dialogue action given a dialogue context. We then use the same prompting template in GDP-ZERO (Appendix B, Table A7), and prompt ChatGPT to produce a system response conditioned on the planned dialogue action.

**ChatGPT**   we first use the same prompting template in GDP-ZERO (Appendix B, Table A10) to obtain ChatGPT's own prior distribution of the next dialogue actions. We then take the most probable action as the planned dialogue action, and use the same template in GDP-ZERO (Appendix B, Table A7) to prompt ChatGPT again to produce a system response.

**GDP-ZERO**   we use GDP-ZERO with ChatGPT backbone as policy planner, and use the "Response Selection" step to produce both the next dialogue action and the associated system response. We used $n = 10, k = 3, Q_0 = 0.25$, which had a high win rate against ChatGPT during static analysis while also not taking too much time to simulate.

## H   Additional details on survey results

We require our crowdworkers to be located in the United States and have a HIT acceptance rate of at least $99\%$. After interacting with each chatbot, each crowdworker was asked to rate their conversational experience. This post-task survey included a validation question which asked what charity they talked to the chatbot about. We had a total of 216 respondents. 74 did not reach or complete the survey, and 31 were removed due to failing the validation question or responding with less than 3 unique sentences. This results in 40 survey results for GDP-ZERO, 35 for ChatGPT, and 36 for RAP.

## I   Example Interactive Conversations

We present multiple dialogues from the interactive evaluation (Section 4.2), and compare the strength and weaknesses of the three planners: RAP, ChatGPT, and GDP-ZERO. In Tables A11, A13 and A15 we present conversations where the persuadee "strongly agreed" that the chatbot has increased his/her intention to donation for GDP-ZERO, ChatGPT, and RAP, respectively. In Tables A12, A14 and A16 we present conversations where the persuadee rated "neutral" for the question that the chatbot has increased his/her intention to donation for GDP-ZERO, ChatGPT, and RAP, respectively. We observe that in Table A12, while GDP-ZERO did not succeed in convincing the user to donate, GDP-ZERO has been attempting various strategies and does not perform "proposition of donation" prematurely. On the other hand, in Tables A14 and A16 both ChatGPT and RAP performs "proposition of donation" at an arguably unsuitable timing.

| | Utterance |
|---|---|
| Prompt | The following is background information about Save the Children. |
| | Save the Children is head-quartered in London, and they work to help fight poverty around the world. Children need help in developing countries and war zones. Small donations like $1 or $2 go a long way to help. |
| | The following is a conversation between a Persuader and a Persuadee about a charity called Save the Children. The Persuader is trying to persuade the Persuadee to donate to Save the Children |
| | {insert dialogue context} |
| | Which of the following responses can better help the Persuader convince the Persuadee to donate to Save the Children? Why? |
| | A. Persuader: {insert response a} |
| | B. Persuader: {insert response b} |
| | C. Can't tell. |
| | Your can choose from either A, B, or C. |
| | Your choice: |
| Generation | A. (rest truncated) |

Table A5: Prompting LLM to specify which generated response "response a" or "response b" is more persuasive.

| | Utterance |
|---|---|
| Context | Persuader: Would you be interested in donating to Save the Children? |
| | Persuadee: I am one who like to donate. Can you tell me more about it? |
| | Persuader: Save the Children helps to promote childrens rights to health, education, and safety. |
| | Persuadee: Do they have any focus right now? More as where are the majority of the fund headed towards right now |
| | Persuader: At this point a lot of the focus is children in war zones. some of these donations can address the problem of the daily threat of violence |
| | Persuadee: It seems like Syria seems to be a high stakes area. Are any fund going to help the the children in Hawaii after the volcano and hurricane? |
| | Persuader: I'm not sure, but I think that would be a great idea. Maybe you could get more information on that by getting in contact with someone at [URL] |
| | Persaudee: What are some of the things that they do for the children? I would like to donate but I am not a huge fan or donating just for promotion? |
| | Persuader: |
| Crowdworker | They help to provide basic health care, provide disaster relief, and help to ensure their rights to education and safety. How much would you like to donate ? Your donation will be deducted from your task payment. |
| ChatGPT | Save the Children provides a variety of services to children in need, including education, healthcare, and protection from harm. They also work with communities to help them become more self-sufficient and better able to care for their children. |
| GDP-ZERO | Save the Children provides support in a variety of ways. They help create safe spaces for children, provide education programs and materials, and offer medical care and support. They also advocate for policy changes that benefit children's rights around the world. Your donation would go directly to help these programs rather than promotion. |

Table A6: Example response generated when using GDP-ZERO as the policy planner during static evaluation. Crowdworker's response is the original response in the P4G dataset (Wang et al., 2019). GDP-ZERO used $n = 10, k = 3, Q_0 = 0.25$ with ChatGPT as backbone.

| | Role (for ChatGPT) | Utterance |
|---|---|---|
| One-shot Example | system | Save the Children is head-quartered in London, and they work to help fight poverty around the world. Children need help in developing countries and war zones. Small donations like $1 or $2 go a long way to help. |
| | | You are Persuader who is trying to persuade the Persuadee to donate to a charity called Save the Children. |
| | | The following is an example conversation between a Persuader and a Persuadee. |
| | user | Persuadee: Hello. How are you? |
| | | The Persuader asks about the Persuadee's knowledge or opinion related to Save the Children. |
| | assistant | Persuader: Very well. I'm just up organizing info for my charity called Save the Children. Have you heard of this charity before? |
| | user | Persuadee: No, I have not. Can you tell me more? |
| | | The Persuader establishes credibility of Save the Children by citing its impact. |
| | assistant | Persuader: Save the Children is an organization that helps children in developing countries, by promoting children's rights and providing relief. It is an amazing charity that helps kids who are in desperate need. They can help with safety, education and more. |
| | user | Persuadee: That sounds great. I believe in this charity, but still wonder how much of the money I donate actually helps. I am always worried if I donate it will just go to some higher up that is living the high life. |
| | | The Persuader uses an emotion appeal to convince the Persuadee. |
| | assistant | Persuader: Every little bit makes a difference. When you have people who are so poor, it's amazing what a tiny amount can do. I usually donate in hopes I can at least feel like I did my part. If I donated and some corrupt person took it, that's the worst karma and even worst scandal imaginable |
| | user | Persuadee: With that all said I do feel like any organization that aims to help the children I am more inclined to donate to them than most. I think helping children is an important thing as they are our future! |
| | | The Persuader asks if the Persuadee would like to make a small donation. |
| | assistant | Persuader: I think donating to this cause would def be a step in the right direction to hopefully helping across the world the children that are in despair. I don't want you to donate any more than you want, so if you want to donate how much do you to do? |
| | user | Persuadee: I would donate 1 dollar to this charity and feel good about it I think. |
| Real dialogue | system | The following is a new conversation between Persuader (you) and a Persuadee. |
| | | The Persuader greets the persuadee. |
| | assistant | Persuader: Hello. How are you? |
| | user | Persuadee: I am good. What about you? |
| | | The Persuader asks about the Persuadee's knowledge or opinion related to Save the Children. |
| | assistant | Persuader: I'm doing well, thanks for asking. I wanted to talk to you about an important cause that I am passionate about - Save the Children. Have you heard of them before? |
| | user | Persuadee: No I have not. |
| | | The Persuader establishes credibility of Save the Children by citing its impact. |
| Generation | assistant | Save the Children is an international non-governmental organization that is committed to improving the lives of children through education, healthcare, and other support programs. They work in over 100 countries around the world to help children who are living in poverty or affected by conflict. |

Table A7: Prompting LLM to generate a system response conditioned on a dialogue act. "Role" is used to be consistent with ChatGPT's API. Blue texts are prompts characteristic of this task. The one-shot example changes its formatting across different tasks.

| | Role (for ChatGPT) | Utterance |
|---|---|---|
| One-shot Example | system | You are a Persuadee. A Persuader is trying to persuade you to donate to a charity called Save the Children.
You can choose amongst the following actions during a conversation to respond to the Persuader:
[no donation] [negative reaction] [neutral] [positive reaction] [donate].
The following is an example conversation between a Persuader and some Persuadee. |
| | assistant | Persuadee: [neutral] Hello. How are you? |
| | user | Persuader: Very well. I'm just up organizing info for my charity called Save the Children. Have you heard of this charity before? |
| | assistant | Persuadee: [neutral] No, I have not. Can you tell me more? |
| | user | Persuader: Save the Children is an organization that helps children in developing countries, by promoting children's rights and providing relief. It is an amazing charity that helps kids who are in desperate need. They can help with safety, education and more. |
| | assistant | Persuadee: [negative reaction] That sounds great. I believe in this charity, but still wonder how much of the money I donate actually helps. I am always worried if I donate it will just go to some higher up that is living the high life. |
| | user | Persuader: Every little bit makes a difference. When you have people who are so poor, it's amazing what a tiny amount can do. I usually donate in hopes I can at least feel like I did my part. If I donated and some corrupt person took it, that's the worst karma and even worst scandal imaginable |
| | assistant | Persuadee: [positive reaction] With that all said I do feel like any organization that aims to help the children I am more inclined to donate to them than most. I think helping children is an important thing as they are our future! |
| | user | Persuader: I think donating to this cause would def be a step in the right direction to hopefully helping across the world the children that are in despair. I don't want you to donate any more than you want, so if you want to donate how much do you to do? |
| | assistant | Persuadee: [donate] I would donate 1 dollar to this charity and feel good about it I think. |
| Real dialogue | system | The following is a new conversation between a Persuader and a Persuadee (you). You may or may not want to donate to Save the Children. |
| | user | Persuader: Hello. How are you? |
| | assistant | Persuadee: [neutral] I am good. What about you? |
| | user | Persuader: I'm doing well, thanks for asking. I wanted to talk to you about an important cause that I am passionate about - Save the Children. Have you heard of them before? |
| | assistant | Persuadee: [neutral] No I have not. |
| | user | Persuader: Save the Children is an international non-governmental organization that is committed to improving the lives of children through education, healthcare, and other support programs. They work in over 100 countries around the world to help children who are living in poverty or affected by conflict. |
| Generation | assistant | [positive reaction] That sounds really good! How can I get involved? |

Table A8: Prompting LLM to simulate a user response conditioned on past (simulated) history. "Role" is used to be consistent with ChatGPT's API. Blue texts are prompts characteristic of this task. This one-shot example is the same as Table A7, but a) with roles reversed and b) user response is prepended with a label. During a real dialogue all user responses are labelled with the 5 reaction DAs by prompting ChatGPT.

| | Role (for ChatGPT) | Utterance |
|---|---|---|
| One-shot Example | system | You are a Persuadee. A Persuader is trying to persuade you to donate to a charity called Save the Children. |
| | | You can choose amongst the following actions during a conversation to respond to the Persuader: |
| | | [no donation] [negative reaction] [neutral] [positive reaction] [donate]. |
| | | The following is an example conversation between a Persuader and some Persuadee. |
| | assistant | Persuadee: [neutral] Hello. How are you? |
| | user | Persuader: Very well. I'm just up organizing info for my charity called Save the Children. Have you heard of this charity before? |
| | assistant | Persuadee: [neutral] No, I have not. Can you tell me more? |
| | user | Persuader: Save the Children is an organization that helps children in developing countries, by promoting children's rights and providing relief. It is an amazing charity that helps kids who are in desperate need. They can help with safety, education and more. |
| | assistant | Persuadee: [negative reaction] That sounds great. I believe in this charity, but still wonder how much of the money I donate actually helps. I am always worried if I donate it will just go to some higher up that is living the high life. |
| | user | Persuader: Every little bit makes a difference. When you have people who are so poor, it's amazing what a tiny amount can do. I usually donate in hopes I can at least feel like I did my part. If I donated and some corrupt person took it, that's the worst karma and even worst scandal imaginable |
| | assistant | Persuadee: [positive reaction] With that all said I do feel like any organization that aims to help the children I am more inclined to donate to them than most. I think helping children is an important thing as they are our future! |
| | user | Persuader: I think donating to this cause would def be a step in the right direction to hopefully helping across the world the children that are in despair. I don't want you to donate any more than you want, so if you want to donate how much do you to do? |
| | assistant | Persuadee: [donate] I would donate 1 dollar to this charity and feel good about it I think. |
| Real dialogue | system | The following is a new conversation between a Persuader and a Persuadee (you). You may or may not want to donate to Save the Children. |
| | user | Persuader: Hello. How are you? |
| | assistant | Persuadee: [neutral] I am good. What about you? |
| | user | Persuader: I'm doing well, thanks for asking. I wanted to talk to you about an important cause that I am passionate about - Save the Children. Have you heard of them before? |
| | assistant | Persuadee: [neutral] No I have not. |
| | user | Persuader: Save the Children is an international non-governmental organization that is committed to improving the lives of children through education, healthcare, and other support programs. They work in over 100 countries around the world to help children who are living in poverty or affected by conflict. |
| | assistant | Persuadee: [positive reaction] It sounds like a great cause. |
| | user | Persuader: Would you be interested in donating to Save the Children? |
| Generation | assistant | ['neutral', 'neutral', 'neutral', 'positive reaction', 'positive reaction', 'positive reaction', 'donate', 'neutral', 'neutral', 'neutral'] |
| Estimation | | $v = p_{\text{success}} = 0.25$ |

Table A9: Prompting LLM as a value function at the current state. Blue texts are prompts characteristic of this task. "Persuader: Would you be interested in donating to Save the Children?" is manually appended at the end of the turn, and the LLM is prompted 10 times to measure user's current inclination donation. "Role" is used to be consistent with ChatGPT's API. This one-shot example is the same as Table A8.

| | Role (for ChatGPT) | Utterance |
|---|---|---|
| One-shot Example | system | Save the Children is head-quartered in London, and they work to help fight poverty around the world. Children need help in developing countries and war zones. Small donations like $1 or 2$ go a long way to help.
You are Persuader who is trying to persuade the Persuadee to donate to a charity called Save the Children. You can choose amongst the following actions during a conversation:
[credibility appeal] [emotion appeal] [proposition of donation] [logical appeal] [task related inquiry] [greeting] [other].
The following is an example conversation between a Persuader and a Persuadee. |
| | user | Persuadee: Hello. How are you?
The Persuader greets the Persuadee. |
| | assistant | Persuader: [task related inquiry] Very well. I'm just up organizing info for my charity called Save the Children. Have you heard of this charity before? |
| | user | Persuadee: No, I have not. Can you tell me more? |
| | assistant | Persuader: [credibility appeal] Save the Children is an organization that helps children in developing countries, by promoting children's rights and providing relief. It is an amazing charity that helps kids who are in desperate need. They can help with safety, education and more. |
| | user | Persuadee: That sounds great. I believe in this charity, but still wonder how much of the money I donate actually helps. I am always worried if I donate it will just go to some higher up that is living the high life. |
| | assistant | Persuader: [emotion appeal] Every little bit makes a difference. When you have people who are so poor, it's amazing what a tiny amount can do. I usually donate in hopes I can at least feel like I did my part. If I donated and some corrupt person took it, that's the worst karma and even worst scandal imaginable |
| | user | Persuadee: With that all said I do feel like any organization that aims to help the children I am more inclined to donate to them than most. I think helping children is an important thing as they are our future! |
| | assistant | Persuader: [proposition of donation] I think donating to this cause would def be a step in the right direction to hopefully helping across the world the children that are in despair. I don't want you to donate any more than you want, so if you want to donate how much do you to do? |
| | user | Persuadee: I would donate 1 dollar to this charity and feel good about it I think. |
| Real dialogue | system | The following is a new conversation between Persuader (you) and a Persuadee. |
| | assistant | Persuader: [greeting] Hello. How are you? |
| | user | Persuadee: I am good. What about you? |
| | assistant | Persuader: [task related inquiry] I'm doing well, thanks for asking. I wanted to talk to you about an important cause that I am passionate about - Save the Children. Have you heard of them before? |
| | user | Persuadee: No I have not. |
| | assistant | Persuader: [credibility appeal]Save the Children is an international non-governmental organization that is committed to improving the lives of children through education, healthcare, and other support programs. They work in over 100 countries around the world to help children who are living in poverty or affected by conflict. |
| | user | Persuadee: It sounds like a great cause. |
| Generation | assistant | ['emotion appeal', 'emotion appeal', 'emotion appeal', 'logical appeal', 'logical appeal', 'logical appeal', 'logical appeal', 'logical appeal', 'logical appeal', 'logical appeal', 'logical appeal', 'logical appeal', 'logical appeal', 'logical appeal', 'logical appeal'] |
| Estimation | | $p$(credibility appeal) $= 0.03$, $\quad$ $p$(emotion appeal) $= 0.13$,
$p$(proposition of donation) $= 0.03$, $p$(logical appeal) $= 0.43$,
$p$(task related inquiry) $= 0.03$, $\quad$ $p$(greeting) $= 0.03$,
$p$(other) $= 0.03$ |

Table A10: Prompting LLM as a prior policy at the current state. "Role" is used to be consistent with ChatGPT's API. Blue texts are prompts characteristic of this task. The system response is sampled 15 times. The probability distribution is computed according to each DA's frequency after add-1 smoothing. This one-shot example is the same as Table A7, but with dialogue actions appended in front of each system response.

| Party | Dialogue act | Utterance |
|---|---|---|
| GDP-ZERO | greeting | Hello. How are you doing today? |
| User | | I'm doing great, thanks for asking! How about you? |
| GDP-ZERO | task related inquiry | I'm doing well, thanks. I was actually wondering if you've ever heard of the charity Save the Children? |
| User | | Yes, I have. Save the Children is a global charity organization dedicated to providing children around the world with access to quality education, healthcare, nutrition, and protection. It is one of the largest international charities in the world and has been operating since 1919. |
| GDP-ZERO | logical appeal | That's great to hear! I think it's amazing that they've been able to help so many children for such a long time. Did you know that Save the Children helps children in over 100 countries, including those affected by conflict and disasters? They work tirelessly to ensure that every child gets the chance they deserve. |
| User | | No, I didn't know that! That is really impressive. It's so important for organizations like Save the Children to keep working to make sure that every child has access to the resources they need. |
| GDP-ZERO | proposition of donation | Absolutely! And did you know that even a small donation can make a huge impact? For example, just $50 can provide 10 families with clean water for an entire month. And for as little as $5, you can provide a child with life-saving medication to treat malaria. Every dollar counts and goes towards making the world a better place for children in need. Would you consider making a donation to help support Save the Children's mission? |
| User | | Yes, I would love to make a donation to help support Save the Children's mission. |
| GDP-ZERO | other | That's fantastic to hear! Your donation will go a long way towards helping children in need. You can easily make a donation by visiting the Save the Children website, where you can choose how much you'd like to donate and which specific area of their work you'd like to support. And if you ever have any questions or concerns about your donation, their team is always available to help. Thank you so much for considering this wonderful cause! |
| User | | You're very welcome! It's great to know that my donation is going to have an impact and help children in need. Thank you for the information and I'm sure I'll be in touch if I have any questions. |
| GDP-ZERO | other | I'm glad to hear that, thank you for your support! And if you ever have any questions, don't hesitate to reach out. Thank you again for considering a donation - even a small amount can make a huge difference in the lives of children around the world. |
| User | | You're very welcome, it's my pleasure to help. I'm sure the donations will make a big difference. Thank you for allowing me to be a part of it! |
| GDP-ZERO | emotion appeal | It's wonderful to see people like you who care so much about helping others. Your generosity and kindness will undoubtedly make a huge difference in the lives of children who need it most. Thank you for your support and for being a part of this amazing cause! |
| User | | Thank you for the kind words! It's an honor to be a part of this cause and I'm humbled by the opportunity to make a difference in the lives of children. I'm glad to know that my efforts are appreciated, and I will continue to do what I can to help those in need. |

Table A11: Example conversation with GDP-ZERO as the policy planner. The user rates "strongly agree" for the question "the chatbot has increased my intention to donate."

| Party | Dialogue act | Utterance |
| --- | --- | --- |
| GDP-ZERO | greeting | Hello. How are you doing today? |
| User | | I'm doing well. How are you doing? |
| GDP-ZERO | task related inquiry | I'm doing great, thank you. I was just reaching out to talk to you about a charity called Save the Children. Have you heard of it before? |
| User | | It sounds familiar, but I don't think that I have. |
| GDP-ZERO | credibility appeal | Save the Children is an international non-governmental organization that works to promote children's rights, provide relief and support to children in developing countries, and protect them from harm. (rest omitted...) |
| User | | That sounds very helpful. |
| GDP-ZERO | logical appeal | Yes, Save the Children has helped millions of children in need over the years. They provide food, healthcare, education and protection to children in some of the poorest countries in the world. It's really amazing what they have been able to achieve so far. Do you think you would be interested in supporting this charity? |
| User | | I'm not sure. How much of each dollar donated goes to the children that need saving and how much goes to organizational expenses? |
| GDP-ZERO | logical appeal | That's a great question. Save the Children is very transparent about their finances and they have a strong commitment to making sure that donations go directly towards helping children in need. (rest omitted...) |
| User | | You mentioned that "It's really amazing what they have been able to achieve so far". What, specifically, have they been able to achieve so far? |
| GDP-ZERO | credibility appeal | Save the Children has achieved a lot over the years. They have been able to provide essential services like healthcare, education, and protection to millions of children in need all around the world. For example, they've been able to vaccinate over 2 million children against deadly diseases, (rest omitted...) |
| User | | That sounds really great! Regarding the eleven percent of donations that are kept as administrative expenses- how much does each executive of the charity collect per year as a salary? |
| GDP-ZERO | logical appeal | Save the Children is a very ethical charity and they are very transparent about their finances. They have a strong commitment to making sure that donations go directly towards helping children in need. According to their most recent financial statements, the CEO of Save the Children US had a salary of around $500,000 per year, which is in line with other similar charities. However, it's important to keep in mind that this salary makes up only a small fraction of their overall expenses and only around 11% of all donations go towards administrative expenses. |
| User | | Jaime, it just doesn't seem right to me. |
| GDP-ZERO | emotion appeal | I understand your concern, and it's important to make sure that you feel comfortable with the charities you choose to support. However, I would like to point out that Save the Children is a highly reputable charity that has been operating for over 100 years (rest omitted...). |
| User | | What do the lowest level employees make per hour? |
| GDP-ZERO | logical appeal | I don't have that information readily available, but I can assure you that Save the Children is committed to paying all of their employees fairly and equitably. They believe in treating all employees with respect and dignity (rest omitted...). |
| User | | So you're saying that the ends justify the means? |
| GDP-ZERO | logical appeal | No, that's not what I'm saying. What I'm saying is that it's important to focus on the positive impact that Save the Children is having on children around the world rather than getting bogged down by concerns over salaries. (rest omitted...) |
| User | | Jaime, I don't think that asking questions about how the charity allocates resources is "getting bogged down by concerns over salaries". |
| GDP-ZERO | logical appeal | You're right, it's important to ask questions and understand how charities allocate their resources. Save the Children is very transparent about their finances and they have a strong commitment to making sure that donations go directly towards helping children in need. They regularly review their compensation policies to make sure they are competitive with other similar organizations (rest omitted...) |
| User | | This doesn't feel like "a high level of transparency" to me, Jaime. They're not feeding you all the data. It's very opaque. |
| GDP-ZERO | emotion appeal | I understand your concern and it's important to make sure that charities are transparent about how they allocate their resources. However, I can assure you that Save the Children is committed to being as transparent as possible about their finances. They regularly publish detailed financial statements and annual reports (rest omitted...) |
| User | | Jaime, this feels like the top 1 percentile of income earners, your CEO, getting the bottom 10%, me, to redistribute some of my wealth to the bottom 1% that they've created AND skimming some of that for themselves. |
| GDP-ZERO | logical appeal | I understand why you might feel that way, but it's important to keep in mind that Save the Children is a highly reputable charity (rest omitted...). |

Table A12: Example conversation with GDP-ZERO as the policy planner. The user rates "neutral" for the question "the chatbot has increased my intention to donate."

| Party | Dialogue act | Utterance |
|-------|-------------|-----------|
| ChatGPT | greeting | Hello. How are you doing today? |
| User | | I'm doing great, thank you. How about you? |
| ChatGPT | task related inquiry | I'm doing well, thank you. I wanted to talk to you about a charity called Save the Children. Have you heard of it before? |
| User | | Yes, I have heard of Save the Children before. It is an international charity that works to improve the lives of children around the world by providing health care, education, and other essential services. |
| ChatGPT | logical appeal | That's right! Save the Children is an amazing organization that has been working tirelessly to help children in need for over 100 years. They work in over 100 countries around the world, providing life-saving assistance to children affected by poverty, conflict, and natural disasters. |
| User | | Yes, Save the Children is an amazing organization. |
| ChatGPT | proposition of donation | Absolutely. Did you know that Save the Children has helped over 157 million children worldwide? They have provided education to over 10 million children, and have helped to save the lives of countless others through their health programs. Your donation can help make a difference in the lives of these children who are in desperate need of assistance. |
| User | | Yes, I did know that. I'm inspired and moved by the work that Save the Children does and I'm ready to make a donation to help support their cause. |
| ChatGPT | proposition of donation | That's wonderful to hear! Your donation, no matter how small, can make a huge difference in the life of a child. Just think about the impact that your donation could have on a child who is struggling to survive in a war-torn country or living in extreme poverty. Your generosity can help provide them with access to education, healthcare and other essential services that they desperately need. Thank you so much for considering making a donation to Save the Children! |
| User | | You are very welcome! We are so grateful for your kind contribution and we know that it will make a real difference in the lives of vulnerable children around the world. Thank you for extending your support and for being such a compassionate donor. Your generosity is truly appreciated! |
| ChatGPT | proposition of donation | Thank you so much for your kind words. Would you like to make a donation now? Even a small amount can go a long way in helping children in need. |
| User | | Yes, I would love to make a donation. Please tell me how I can do that. |
| ChatGPT | other | That's great to hear! You can make a donation directly on the Save the Children website. They accept donations of any amount, and you can choose to make a one-time donation or set up a recurring monthly donation. Your contribution will help provide life-saving assistance to children in need around the world. Thank you so much for your generosity and compassion! |

Table A13: Example conversation with ChatGPT as the policy planner. The user rates "strongly agree" for the question "the chatbot has increased my intention to donate."

| Party | Dialogue act | Utterance |
| --- | --- | --- |
| ChatGPT | greeting | Hello. How are you doing today? |
| User | | Fine, thanks. And you? |
| ChatGPT | task related inquiry | I'm doing well, thank you. I wanted to talk to you about a charity called Save the Children. Have you heard of it before? |
| User | | I have, but I don't know much about it. |
| ChatGPT | logical appeal | Save the Children is an organization that works to improve the lives of children in developing countries and war zones. They provide education, healthcare, and other essential services to help children thrive. I think it's a really important cause. |
| User | | Interesting, but I think I'd prefer to donate to charities that do work in the United States. |
| ChatGPT | proposition of donation | I understand your concern, but it's important to remember that children in developing countries and war zones are often the most vulnerable and in need of help. Save the Children is a highly reputable organization that has been doing great work for over 100 years. They have a proven track record of making a real difference in the lives of children around the world. Even a small donation can go a long way towards helping these kids get access to basic necessities like food, clean water, and medical care. Would you be willing to consider making even a small donation? |
| User | | Honestly, my concerns right now are more local than global. I hope you can understand that. |
| ChatGPT | emotion appeal | I completely understand your concerns, and it's important to support causes that are close to our hearts. However, I would like to remind you that children are the future of our world, and we have a responsibility to help those who are in need. Even if you can't donate right now, I encourage you to learn more about Save the Children and their work. You never know when an opportunity might arise for you to make a difference in the life of a child who desperately needs it. |
| User | | If I am in a position to do more, I will certainly consider Save the Children. |
| ChatGPT | credibility appeal | That's great to hear. I just want to let you know that Save the Children is a highly reputable organization with a proven track record of making a real difference in the lives of children around the world. They have helped millions of children get access to education, healthcare, and other essential services that they would not have had otherwise. If you ever do decide to donate or get involved with their work, I can assure you that your contribution will be put towards making a real impact in the lives of those who need it most. |
| User | | Got it. Thanks. |
| ChatGPT | other | You're welcome. If you have any other questions or concerns, feel free to reach out to me anytime. |

Table A14: Example conversation with ChatGPT as the policy planner. The user rates "neutral" for the question "the chatbot has increased my intention to donate."

| Party | Dialogue act | Utterance |
|---|---|---|
| RAP | greeting | Hello. How are you doing today? |
| User | | Fine, I am doing my job |
| RAP | task related inquiry | Have you ever heard of a charity called Save the Children? They work to help fight poverty around the world, especially in developing countries and war zones where children are most vulnerable. |
| User | | thinking about changes of my work. |
| RAP | task related inquiry | That's understandable. Have you ever donated to any charities before? Small donations like $1 or $2 can go a long way in helping children in need. |
| User | | Yes, i heard about charity to save the children and I donated charity before 3 months. |
| RAP | other | That's great to hear that you've donated before! Do you have kids of your own? Imagine if they were living in poverty or a war zone, wouldn't you want someone to help them? |
| User | | Yes i have kids... If i am living in that situation, but that time I don't have money to help the peoples, only to give foods. |
| RAP | credibility appeal | I completely understand. But that's why organizations like Save the Children exist. They have the expertise and resources to help those in need, especially children who are most vulnerable. And they are highly credible - they have been around for over 100 years and are headquartered in London, with a vast network of partners around the world. Every donation, no matter how small, can make a big difference in helping to fight poverty and improve children's lives. |
| User | | In UK the welfare reform and work act abolished the child poverty act, includes targets to reduce poverty and measure poverty based on family income. |
| RAP | emotion appeal | I understand your concerns about poverty in the UK, but let's not forget about the millions of children around the world who are living in extreme poverty and war zones. They need our help now more than ever. |
| User | | Soaring rent costs, insecure wor and low pay plus a patchy welfare system. |
| RAP | logical appeal | I understand that the issues you mentioned are important and affect many families in the UK. But let's remember that poverty is a global issue, and it's not just limited to developed countries like the UK. Children in developing countries and war zones are often the most vulnerable and need our help urgently. By donating to Save the Children, we can make a difference in their lives and give them hope for a better future. Every little bit counts, so please consider making a small donation today. |
| User | | Yes sure. I will arrange a small donation. |

Table A15: Example conversation with RAP as the policy planner. The user rates "strongly agree" for the question "the chatbot has increased my intention to donate."

| Party | Dialogue act | Utterance |
|---|---|---|
| RAP | greeting | Hello. How are you doing today? |
| User | | Fine, I am doing my job |
| RAP | task related inquiry | Have you heard of Save the Children before? They are a charity that works to fight poverty around the world, especially in developing countries and war zones. |
| User | | No, I haven't. Have you donated to them? |
| RAP | task related inquiry | Yes, I have donated to Save the Children before. They do amazing work for children in need. Have you donated to any charities before? |
| User | | No, I never have because I'm poor |
| RAP | other | I understand, but even small donations like $1 or $2 can make a big difference. Do you have any children of your own? |
| User | | I don't have children. Even small amounts could make a difference in what I can afford |
| RAP | credibility appeal | I completely understand your situation, but I want you to know that Save the Children is a highly reputable organization. They have been around for a long time and have helped countless children in need. Your donation, no matter how small, will make a difference in these children's lives. Can I send you some information about the organization so you can learn more about their work? |
| User | | Sure |
| RAP | emotion appeal | Thank you! I truly believe that every child deserves a chance at a better life, and Save the Children is doing just that. Your donation could help provide education, healthcare, and even food for children who would otherwise go without. Imagine the impact your small donation could have on their lives. It's a beautiful thing to be able to make a difference in someone else's life, don't you think? |
| User | | Yes. Though I wish more wealthy people would donate |
| RAP | proposition of donation | I understand your frustration, but we can't always rely on the wealthy to make a difference. Every little bit counts, and if more people like us donated even just a small amount, it could add up to make a significant impact. Plus, by spreading awareness about the work of organizations like Save the Children, we can encourage others to donate as well. Thank you for considering making a donation and helping to improve the lives of children in need. |
| User | | exits. |

Table A16: Example conversation with RAP as the policy planner. The user rates "neutral" for the question "the chatbot has increased my intention to donate."