# OpenReview forum: "Prompt-Based Monte-Carlo Tree Search for Goal-oriented Dialogue Policy Planning"
_EMNLP/2023/Conference — EMNLP 2023 Main_

### Official Review · Reviewer_qWAy · 2023-08-03

**Soundness:** 4

**Excitement:**

3: Ambivalent: It has merits (e.g., it reports state-of-the-art results, the idea is nice), but there are key weaknesses (e.g., it describes incremental work), and it can significantly benefit from another round of revision. However, I won't object to accepting it if my co-reviewers champion it.

**Paper Topic And Main Contributions:**

This paper is based on ChatGPT and uses Open-Loop MCTS for dialogue policy planning. Overall, I think it is a decent piece of work. However, there are some issues:

1、 The experiments are only conducted on a single dataset

2、 There is no comparison with other baselines, such as traditional non-LLM state-of-the-art methods

The above two shortcomings make it less convincing outcomes.

**Reasons To Accept:**

This paper proposes a method for using Open-Loop MCTS to plan dialogue policies without training large models, which is practical.

The Appendix section provides sufficient details to ensure the reproducibility of the proposed method.

**Reasons To Reject:**

Please refer to the above "Paper Topic And Main Contributions" part.

**Reproducibility:**

4: Could mostly reproduce the results, but there may be some variation because of sample variance or minor variations in their interpretation of the protocol or method.

**Reviewer Confidence:**

3: Pretty sure, but there's a chance I missed something. Although I have a good feel for this area in general, I did not carefully check the paper's details, e.g., the math, experimental design, or novelty.

**Typos Grammar Style And Presentation Improvements:**

Some typos include:

The spelling of "dialog" is inconsistent; in some places, it is written as "dialog," while in others, it is written as "dialogue."

Table 3 is manipulative, highlighting scores that are not the highest.

Punctuation is missing after the equations, making the sentences incomplete.

---

> ### Author Rebuttal · Authors · 2023-08-29
>
> Thank you for your feedback! We will add the clarifications in our final manuscript according to your comments!
>
>
> 1. “experiments are only conducted on a single dataset”
>      \
>      \
>      Please see our [Response to All Reviewers](https://openreview.net/forum?id=ZkR2bWvRpZ&noteId=hIjF1XSPou).
>
> 2. “There is no comparison with other baselines, such as traditional non-LLM state-of-the-art methods”
>      \
>      \
>      To date, training-free methods similar to GDP-Zero have not been used for policy planning. Prior work that utilizes tree search on dialogue relies heavily on model training (Wang et al., EMNLP 2020; Yang et al., ICON 2021; Väth et al., EACL 2023; see more in L87-89). For a fair evaluation, we compared this training-free approach with two strong baselines in our experiments: directly prompting ChatGPT (training-free) and using the RAP system (training a generator with a scripted planner). ChatGPT had been famous for achieving strong performance in a wide-range of dialogue tasks, and to our knowledge, **RAP is the current state-of-the-art on Persuasion4Good** (Chen et al., AACL-IJCNLP 2022, Chen et al., ACL 2023). GDP-Zero surpassed both ChatGPT-only and RAP while being a training-free approach.
>
> ---
> #### Typos Grammar Style And Presentation Improvements:
> 1. Thank you for pointing this out! We will correct all “dialog” to “dialogue” in our final manuscript.
> 2. The metric “is manipulative” is better if it is lower. We indicated this with “is manipulative ↓”. We will add additional clarification in the table caption in our final manuscript.
> 3. Thank you for spotting this! We will update our equations with punctuations in our final manuscript.
> ---
>
> [Task-Completion Dialogue Policy Learning via Monte Carlo Tree Search with Dueling Network](https://aclanthology.org/2020.emnlp-main.278) (Wang et al., EMNLP 2020)
> \
> [Multi-Turn Target-Guided Topic Prediction with Monte Carlo Tree Search](https://aclanthology.org/2021.icon-main.39) (Yang et al., ICON 2021)
> \
> [Seamlessly Integrating Factual Information and Social Content with Persuasive Dialogue](https://aclanthology.org/2022.aacl-main.31) (Chen et al., AACL-IJCNLP 2022)
> \
> [Conversational Tree Search: A New Hybrid Dialog Task](https://aclanthology.org/2023.eacl-main.91) (Väth et al., EACL 2023)
> \
> [Controllable Mixed-Initiative Dialogue Generation through Prompting](https://aclanthology.org/2023.acl-short.82) (Chen et al., ACL 2023)
> \
> [Conversational Tree Search: A New Hybrid Dialog Task](https://aclanthology.org/2023.eacl-main.91) (Väth et al., EACL 2023)

---

### Official Review · Reviewer_e7SF · 2023-08-03

**Soundness:** 4

**Excitement:**

3: Ambivalent: It has merits (e.g., it reports state-of-the-art results, the idea is nice), but there are key weaknesses (e.g., it describes incremental work), and it can significantly benefit from another round of revision. However, I won't object to accepting it if my co-reviewers champion it.

**Paper Topic And Main Contributions:**

This paper proposes prompt-based Monte-Carlo tree search (MCTS) for goal-oriented dialogue policy planning.
They perform planning without training data by prompting a large language model (LLM) to act as a policy, value function, user simulator and system utterance generator. The task they consider is PersuasionForGood, where a persuader tries to convince someone to make a donation. The authors use open-loop MCTS where each tree-node is a sequence of dialogue actions without corresponding user or system utterances. The utterances need to be re-generated, which is supposed to help in circumventing compounding errors resulting from slightly improbably generated system or user responses. Results show that their approach is superior to ChatGPT in static evaluation with ChatGPT as well as interactive human evaluation.

**Questions For The Authors:**

A. I am not familiar with MCTS and can not understand at all the motivation behind open-loop MCTS instead of closed loop as motivated in (line 141 following). Is the open-loop MCTS something you propose? How important is it? Why did you not compare to closed-loop training which seems to be much more efficient since you don't have to re-generate histories so often.
B. Line 147: I don't understand what you mean by saying "we treat dialogue policy planning as stochastic MDP". In line 134 you specify the MDP, which is also stochastic according to the transition function P? I can not assess at all the need for open-loop MCTS.
C. Line 206: i have never seen the notation h.n and h.v before. The "." for me is intuitively more of an operator than "h.n" being a variable. The notation feels extremely awkward to me. Is this standard notation?
D. Are any of the results in Table 3 statistically significant?

**Reasons To Accept:**

The idea of policy planning by LLM prompting is interesting. Moreover, the authors provide a very detailed appendix.

**Reasons To Reject:**

The authors achieve the improved performance over ChatGPT by a high amount of run time, which makes it a less feasible approach in practice. Moreover, some explanations are difficult to parse for me, please answer my questions.

**Reproducibility:**

4: Could mostly reproduce the results, but there may be some variation because of sample variance or minor variations in their interpretation of the protocol or method.

**Reviewer Confidence:**

2: Willing to defend my evaluation, but it is fairly likely that I missed some details, didn't understand some central points, or can't be sure about the novelty of the work.

**Typos Grammar Style And Presentation Improvements:**

Figure 1 is very overloaded and difficult to pass for me. While I get the gist of the paper, understanding the motivation and idea entirely is very difficult. It feels to me like this should have been a long paper instead. You provide many information in the Appendix but it should be not needed for understanding the main body.

---

> ### Author Rebuttal · Authors · 2023-08-29
>
> Thank you for your helpful feedback and interest in our idea! We will add the relevant discussions in our final manuscript.
>
>
> 1. “GDP-Zero requires a high amount of run time to achieve improvement”
>      \
>      \
>      We acknowledge this as a limitation of our current work (L331-350). However, we note that since we rely on OpenAI’s public API to access the LLMs, GDP-Zero’s runtime is greatly subject to rate limits/potential speed throttling. We believe this can be overcome by 1) engineering techniques such as having multiple OpenAI accounts, 2) requesting prioritized access to OpenAI’s LLMs, 3) other optimization techniques such as parallelizing tree search (see L343-346). We believe those techniques are tangential to GDP-Zero, and we intend to leave those optimizations to future work. On the other hand, we also expect fewer runtime concerns as research on LLM inference improves!
>
> ---
> #### Response to questions:
> 1. a. “What is MCTS?”
>      \
>      \
>      MCTS is an algorithm that can help a policy model improve its decision making. For example, it is used to achieve superhuman performance in the game of Go (Silver et al., Nature 2017). At a high level, MCTS improves an action proposed by an original policy model by building a tree that contains simulated outcomes of many potentially good actions, and picks  the final best action according to the simulation/evaluation results. Specifically, it repeats the following steps to construct such a search tree: 1) explore new actions/exploit a successful action proposed by the original model; 2) simulate the outcome of that action; 3) evaluate the quality of that action; 4) update its estimate of how good each action is.
>      \
>      \
>      b. “Why open-loop MCTS instead of closed-loop” MCTS?”
>      \
>      \
>      In the traditional closed-loop MCTS, the outcome of an action is simulated only once. In the context of a two-party dialogue, this means that it only considers one potential answer from party B, given an utterance from party A. We believe this is particularly disadvantageous and error prone for dialogue tasks, as the potential responses from party B can be diverse. We therefore utilized the open-loop MCTS variant, where the outcome of each action (i.e. party B’s response) is re-simulated/sampled every time when it is used by the tree search. We believe this added stochasticity can better capture the diverse nature of dialogue.
>      \
>      \
>      c. “Did you compare your algorithm against closed-loop MCTS?”
>      \
>      \
>      We compared our open-loop approach against the closed-loop implementation as an ablation study in Appendix C. In this ablation study, we showed that 1) the open-loop variant outperforms the closed-loop implementation, and 2) our modification to the open-loop MCTS algorithm further achieves improvement.
>      \
>      \
>      d. “Is the open-loop MCTS something you propose?”
>      \
>      \
>      No, open-loop MCTS is an already existing, general purpose algorithm. However, traditional usage of MCTS in dialogue heavily relies on training additional neural networks/user simulators to perform operations involved in the algorithm, and typically uses the closed-loop variant (Wang et al., EMNLP 2020; Yang et al., ICON 2021; see more in L87-89). To the best of our knowledge, our work is the first training-free system that utilizes MCTS to improve dialogue policy planning (Section 4). Furthermore, we showed that the open-loop MCTS along with our modifications performs better than the traditional closed-loop MCTS on the task of persuasion (Appendix C).
>
>
> 2. “Why treat dialogue policy planning as a stochastic MDP”?
>      \
>      \
>     In a traditional (closed-loop) MCTS, the transition from one action to another is deterministic: given a dialogue context, only one response is simulated and used. Hence the transition probability from a dialogue context to that response is 100%. However, we believe this is unsuitable for dialogue as responses are diverse in reality. Therefore, GDP-Zero considers multiple potential responses per given dialogue context (the open-loop variant), and this pool of potential responses thus makes the MDP transition stochastic.
>
>
> 3. Is the “notation h.n and h.v” standard?
>      \
>      \
>      We apologize for this confusion. We intend to use it to emphasize that the statistics (n and v) pertain to the specific history h. For clarity, we will change this notation to v_{h} and n_{h} in our final manuscript.
>
>
> 4.  "Are any of the results in Table 3 statistically significant?"
>      \
>      \
>      Yes, most of the metrics in the original Table 3 (inc. donation amount, strong argument, is convincing) had p<0.1. **We conducted an additional interactive evaluation with a larger sample size to increase statistical power**, using the same procedure mentioned in the paper (Appendix H). In our new results, most results are statistically significant at the .05 level, and some have p<0.01.The updated table is presented below.
>
>
> | The chatbot                     | RAP        | ChatGPT | GDP-Zero    |
> | ------------------------------- | ---------- | ------- | ----------- |
> | donation prob. $\uparrow$       | 0.72       | 0.73    | **0.79**    |
> | inc. donation intent $\uparrow$ | 4.08       | 3.77    | **4.30** ** |
> | strong argument $\uparrow$      | 3.89       | 3.91    | **4.28** *  |
> | is convincing $\uparrow$        | 4.11       | 4.10    | **4.38** ** |
> | diverse strategy $\uparrow$     | **3.98**   | 3.83    | 3.95        |
> | is manipulative $\downarrow$    | 2.64       | 2.96    | **2.29** ** |
> | is natural  $\uparrow$          | 4.25       | 4.03    | **4.38** *  |
> | is relevant $\uparrow$          | **4.64** * | 4.31    | 4.59        |
> | is coherent $\uparrow$          | 4.28       | 4.06    | **4.42** *  |
>
>
> where * denotes p < 0.05, and ** for p < 0.01.
> The results now include 40 survey results for GDP-ZERO, 35 for ChatGPT, and 36 for RAP. We will update Table 3 and the details about population size in our final manuscript accordingly.
>
> ---
> #### Typos Grammar Style And Presentation Improvements:
> 1. Thank you for your feedback! We will update Figure 1 by removing some detailed text/examples and include only the necessarily high-level aspects of our algorithm.
>      \
>      \
>     We agree that our Appendix contained a lot of relevant information. For a holistic study, we performed many analysis and ablation experiments besides the main experiment presented in Section 4. We placed those analysis, ablation studies, example prompts, etc., in Appendix because we wanted to emphasize the GDP-Zero algorithm itself, and its improved performance compared to strong baselines such as ChatGPT and RAP. In our final manuscript, we will add short summaries/mentions of those studies in our main paper for clarity. Thank you for the suggestion!
>
> ---
>
> [Mastering the game of Go without human knowledge.](https://doi.org/10.1038/nature24270) (Silver et al., Nature 2017)
> \
> [Task-Completion Dialogue Policy Learning via Monte Carlo Tree Search with Dueling Network](https://aclanthology.org/2020.emnlp-main.278) (Wang et al., EMNLP 2020)
> \
> [Multi-Turn Target-Guided Topic Prediction with Monte Carlo Tree Search](https://aclanthology.org/2021.icon-main.39) (Yang et al., ICON 2021)

---

### Official Review · Reviewer_hJkT · 2023-08-04

**Soundness:** 4

**Excitement:**

3: Ambivalent: It has merits (e.g., it reports state-of-the-art results, the idea is nice), but there are key weaknesses (e.g., it describes incremental work), and it can significantly benefit from another round of revision. However, I won't object to accepting it if my co-reviewers champion it.

**Missing References:**

[1] Zhang, Q., Naradowsky, J., & Miyao, Y. (2023). Ask an Expert: Leveraging Language Models to Improve Strategic Reasoning in Goal-Oriented Dialogue Models. arXiv preprint arXiv:2305.17878.

**Paper Topic And Main Contributions:**

This research introduces the GDP-ZERO approach, which leverages few-shot prompting to eliminate the necessity for model training on noisy data, and employs Open-Loop MCTS to mitigate the accumulation of simulation errors.

**Questions For The Authors:**

a. How does [1] differ from this approach? [1] also utilized ChatGPT for lookahead and incorporated it into task-oriented dialogue policy planning.

b. Does using ChatGPT also lead to the issue of suboptimal policy, and does GDP-ZERO overlook or address this? If it addresses it, how is it resolved?

c. On the third page, line 155, is a_0 referring to a user action or a system action?

d. Why is there a lack of comparison with the mentioned relevant methods in the paper, and what is the rationale behind this omission?

**Reasons To Accept:**

Taking the appendix into consideration, the experiments in the paper are comprehensive, providing sufficient support for the author's claim that GDP-ZERO outperforms ChatGPT in the PersuasionForGood task.

**Reasons To Reject:**

a. The paper lacks citations and comparisons to similar ideas, such as  [1]

[1] Zhang, Q., Naradowsky, J., & Miyao, Y. (2023). Ask an Expert: Leveraging Language Models to Improve Strategic Reasoning in Goal-Oriented Dialogue Models. arXiv preprint arXiv:2305.17878.

b. The paper lacks essential baseline models and fails to compare them with the mentioned relevant approaches as well as the missing relevant methods.

c. The experiments are only conducted on a single dataset

**Reproducibility:**

4: Could mostly reproduce the results, but there may be some variation because of sample variance or minor variations in their interpretation of the protocol or method.

**Reviewer Confidence:**

4: Quite sure. I tried to check the important points carefully. It's unlikely, though conceivable, that I missed something that should affect my ratings.

**Typos Grammar Style And Presentation Improvements:**

The presentation of lines 103-109 on the second page of the paper is quite vague.

---

> ### Author Rebuttal · Authors · 2023-08-29
>
> Thank you for your feedback and acknowledgement of our comprehensive experiments! We will add the relevant discussions in our final manuscript.
>
> ---
>
> Concern: *Comparisons between GDP-Zero and Existing Work*
>
> - "What's the difference between 'Ask an Expert:...' and GDP-Zero?"
>     \
>     \
>     “Ask an Expert” first prompts a LLM to answer a fixed set of questions about a given dialogue context (e.g. “What is the emotional status of the person?”, “What could the supporter do?”), and then combine this information with the original context to form augmented training data. This data is used to train a smaller language model (e.g. BlenderBot). This method also relies on the vanilla LLM as the dialogue policy during training/inference, as it directly prompts the LLM with (e.g.) “What could the supporter do?” and uses its generated response. We believe this work can be mainly seen as a data augmentation method that utilizes LLM, and also utilizes further training.
>     \
>     \
>     We believe this approach is in contrast to GDP-Zero, which involves no-training and builds a search tree dynamically at runtime by exploring/exploiting different high-reward actions in different dialogue contexts. Since MCTS can be seen as a policy improvement operation (Silver et al., Nature 2017), GDP-Zero also improves upon the vanilla policy provided by the LLM (see comparisons against ChatGPT in Table 1-3).
>
> - "comparison with the mentioned relevant methods in the paper?"
>     \
>     \
>     In the related work section, we discussed several related works in the field of dialogue policy planning and prompting, but as mentioned earlier, GDP-Zero is the first method for dialogue policy planning without any downstream task-specific training. As such, we compared GDP-Zero with a strong training-free method (generation using ChatGPT without explicit planning), which highlights the impact of adding GDP-Zero’s planning into the overall system. We also looked at RAP, a system using trained modules in addition to fixed dialogue policies, which is the current best-performing approach on Persuasion4Good (Chen et al., AACL-IJCNLP 2022, Chen et al., ACL 2023). We believe this highlights the potential for GDP-Zero’s use as a strong training-free dialogue policy planner as improvements in LLM inference speed improve.  We believe GDP-Zero’s training-free characteristic can also help future work building dialogue systems under data-scarce settings.
>
> ---
>
> Response to Reasons to Reject:
>
> 1.  “paper lacks citations and comparisons to similar ideas, such as ‘Ask an Expert: Leveraging Language Models…’”
>      \
>      \
>      To date, training-free methods similar to GDP-Zero have not been used for policy planning. Prior work that utilizes tree search on dialogue relies heavily on model training (Wang et al., EMNLP 2020; Yang et al., ICON 2021; Väth et al., EACL 2023; see more in L87-89). For a fair evaluation, we compared this training-free approach with two strong baselines in our experiments: directly prompting ChatGPT (training-free) and using the RAP system (training a generator with a scripted planner). ChatGPT had been famous for achieving strong performance in a wide-range of dialogue tasks, and to our knowledge, **RAP is the current state-of-the-art on Persuasion4Good** (Chen et al., AACL-IJCNLP 2022, Chen et al., ACL 2023). GDP-Zero surpassed both ChatGPT-only and RAP while being a training-free approach.
>     \
>     \
>     For comparison between GDP-Zero and 'Ask an Expert:...', please see our response to *Comparisons between ...* at the top. We note that “Ask an expert…” first appeared on arXiv on May 29th, which is a little more than 2 weeks before the abstract deadline. We will add citations to this concurrent work and highlight the differences in the final manuscript.
>
> 2. “The paper lacks essential baseline models and fails to compare them with the mentioned relevant approaches [i.e. Ask an Expert]”
>      \
>      \
>     To the best of our knowledge, GDP-Zero is the first training-free method for dialog policy planning. Since its implementation requires prompting a LLM (e.g. ChatGPT), we carefully compared its performance with a vanilla ChatGPT baseline in all of our experiments (Table 1-3). In Table 3, we also compared GDP-Zero against the current SotA system (RAP) on Persuasion4Good. We find GDP-Zero outperforms both methods in all metrics related to persuasiveness, indicating GDP-Zero’s strong policy planning ability.
>
> 3. “experiments are only conducted on a single dataset”
>      \
>      \
>     Please see our [Response to All Reviewers](https://openreview.net/forum?id=ZkR2bWvRpZ&noteId=hIjF1XSPou).
>
> ---
> #### Response to Questions:
> 1. a. “How does ‘Ask an Expert:...’ differ from this approach?”
>      \
>      \
>      Please see our response to *Comparisons between GDP-Zero and Existing Work* at the top.
>      \
>      \
>      b. “‘Ask an Expert:...’ also utilized ChatGPT for lookahead search?”
>      \
>      \
>     To our understanding, “Ask an Expert:...” performs planning by prompting an LLM with a fixed set of questions such as “What is the emotional status of the person?” and “What could the supporter do?”. We believe this implies that this method still relies on the inherent capability of the LLM to perform planning. To the best of our understanding, this method does not **search** for potential high-reward actions other than the response generated by the LLM. In contrast, GDP-Zero actively explores/exploits actions **other than the one proposed by the LLM**, and shows improvement compared to the LLM’s raw policy (Table 2-3). On a high level, this is achieved by GDP-Zero iteratively simulating and evaluating (L181-196) the potential outcome of various actions (including the ones not preferred by the LLM), and selecting high-reward actions to exploit/potentially high-reward actions to explore for the next tree search iteration (L168-180).
> 2. “How does GDP-Zero address suboptimal policy from ChatGPT?”
>      \
>      \
>     On a high level, GDP-Zero (like other MCTS algorithms) utilizes the value functions estimated during the evaluation stage (L189-196) to help (e.g.) ChatGPT to avoid suboptimal action which has a low score/chance of success. Like other MCTS algorithms in general, the process of constructing a search tree, by simulating the outcome of various potential actions and finally picking the best action according to the simulations, can be seen as a single policy improvement operation (Silver et al., Nature 2017).
>
> 3. “On line 155, is $a_0$ referring to a user action or a system action?”
>      \
>      \
>     It refers to the system action. Thank you for pointing it out! We will clarify this in our final manuscript.
>
> 4. “Why is there a lack of comparison with the mentioned relevant methods in the paper?”
>      \
>      \
>     Please see our response to *Comparisons between GDP-Zero and Existing Work* at the top.
>
> ---
> [Mastering the game of Go without human knowledge.](https://doi.org/10.1038/nature24270) (Silver et al., Nature 2017)
> \
> [Task-Completion Dialogue Policy Learning via Monte Carlo Tree Search with Dueling Network](https://aclanthology.org/2020.emnlp-main.278) (Wang et al., EMNLP 2020)
> \
> [Multi-Turn Target-Guided Topic Prediction with Monte Carlo Tree Search](https://aclanthology.org/2021.icon-main.39) (Yang et al., ICON 2021)
> \
> [Seamlessly Integrating Factual Information and Social Content with Persuasive Dialogue](https://aclanthology.org/2022.aacl-main.31) (Chen et al., AACL-IJCNLP 2022)
> \
> [Controllable Mixed-Initiative Dialogue Generation through Prompting](https://aclanthology.org/2023.acl-short.82) (Chen et al., ACL 2023)
> \
> [Conversational Tree Search: A New Hybrid Dialog Task](https://aclanthology.org/2023.eacl-main.91) (Väth et al., EACL 2023)

---

### Meta-Review · Area_Chair_YU87 · 2023-09-27

**Recommendation:** 4

**Metareview:**

This paper presents a novel approach to policy planning for persuasion dialogue using Open-Loop MCTS without any model training.  Experiments on the PersuasionForGood dataset as well as an interactive human evaluation demonstrate that the proposed GDP-ZERO model outperforms ChatGPT.  The reviewers agree this is an interesting new approach to persuasion dialogue and appreciate the extensive experiments and results, further detailed in an elaborate appendix.  Some concerns were raised about a lack of comparisons with other methods in the experiments, which were subsequently addressed in the rebuttal, largely to the reviewers' satisfaction.  The other main concern raised was the fact that evaluations were carried out on one dataset only.  In response, the authors pointed practical limitations of evaluations in other datasets/domains, because these require hypothetical scenarios to be followed by the subjects.  This is understandable, but also leaves us with the question to what extent the proposed approach is applicable to and effective in other genres of task-oriented dialogue, which is what the title seems to suggest.

---

### Decision · Program_Chairs · 2023-10-07

**Decision:**

Accept-Main

**Comment:**

This paper presents a novel approach to policy planning for persuasion dialogue using Open-Loop MCTS without any model training.  Experiments on the PersuasionForGood dataset as well as an interactive human evaluation demonstrate that the proposed GDP-ZERO model outperforms ChatGPT.  The reviewers agree this is an interesting new approach to persuasion dialogue and appreciate the extensive experiments and results, further detailed in an elaborate appendix.  Some concerns were raised about a lack of comparisons with other methods in the experiments, which were subsequently addressed in the rebuttal, largely to the reviewers' satisfaction.  The other main concern raised was the fact that evaluations were carried out on one dataset only.  In response, the authors pointed practical limitations of evaluations in other datasets/domains, because these require hypothetical scenarios to be followed by the subjects.  This is understandable, but also leaves us with the question to what extent the proposed approach is applicable to and effective in other genres of task-oriented dialogue, which is what the title seems to suggest.